



# Extending the CW3E Atmospheric River Scale to the Polar Regions

Zhenhai Zhang[1], F. Martin Ralph[1], Xun Zou[1], Brian Kawzenuk[1], Minghua Zheng[1],
Irina V. Gorodetskaya[2], Penny M. Rowe[3], and David H. Bromwich[4]

[1] Center for Western Weather and Water Extremes, Scripps Institution of Oceanography,
University of California San Diego, La Jolla, CA, USA

[2] CIIMAR | Interdisciplinary Centre of Marine and Environmental Research, University of Porto,
Porto, Portugal

[3] NorthWest Research Associates, Redmond, WA, USA

[4] Polar Meteorology Group, Byrd Polar and Climate Research Center, The Ohio State University,
Columbus, OH, USA

Correspondence: Zhenhai Zhang (zhz422@ucsd.edu)



**Abstract.** Atmospheric rivers (ARs) are the primary mechanism for transporting water vapor from low latitudes to polar regions, playing a significant role as drivers of extreme weather, such as heavy precipitation and heat waves in both the Arctic and Antarctica. With the rapidly growing interest in polar ARs during the past decade, it is imperative to establish an objective framework to quantify the strength and impact of these ARs for both scientific research and

practical application. The AR scale introduced by Ralph et al. (2019) ranks ARs based on the duration of AR conditions and the intensity. However, the thresholds of integrated water vapor transport (IVT) used to rank ARs are selected based on the IVT climatology at middle latitudes. These thresholds are insufficient for polar regions due to the substantially lower temperature and moisture content. In this study, we analyze the IVT climatology in polar regions, focusing on the

coasts of Antarctica and Greenland. Then we introduce an extended version of the AR scale tuned to polar regions by adding lower IVT thresholds of 100, 150, and 200 kg m$^{-1}$ s$^{-1}$ to the standard AR scale, which starts at 250 kg m$^{-1}$ s$^{-1}$. The polar AR scale is utilized to examine AR frequency, seasonality, trends, and associated precipitation and surface melt over the Antarctic and Greenland coasts. The polar AR scale better characterizes the strength and impacts of ARs in

the Antarctic and Arctic regions, and has the potential to enhance communications across observation, research, and forecasts for polar regions.





## 1. Introduction

An atmospheric river (AR) is a "long, narrow, and transient corridor of strong horizontal water
vapor transport in the lower troposphere" (American Meteorological Society 2018; Ralph et al.
2018), and it is usually associated with a low-level jet ahead of the cold front of an extratropical
cyclone. ARs are the main mechanism for transporting water vapor from the tropics and
subtropics to the middle and high latitudes (Zhu and Newell 1998; Ralph et al., 2004; Newmann
et al., 2012). At the same time, ARs play a critical role in regional precipitation and flooding in
coastal regions worldwide, including the United States (Guan et al., 2010; Dettinger 2011; Ralph
et al., 2013; Debbage et al., 2017; DeFlorio et al., 2024), Europe (Lavers and Villarini 2013;
Ionita et al., 2020), New Zealand, (Shu et al., 2021; Prince et al., 2021), and South America
(Viale et al., 2018). ARs also contribute to other extreme weather events. They are directly
linked to extreme winds over most coastal regions in the world (Waliser and Guan 2017). ARs
are closely associated with extratropical cyclones, and antecedent AR conditions can provide
extra water vapor inflow to significantly enhance the diabatic process and intensify cyclone
deepening, especially for explosive extratropical cyclones (Zhang and Ralph 2021; Zhang et al.,
2019; Eiras-Barca et al., 2018).

Recent studies have found that ARs have large impacts on regional climate and extreme weather
events in polar regions and exert notable influence on the polar ice. As a prominent supplier of
moisture and heat, ARs often couple with low-level jets, having the potential to induce extreme
hot spells and heatwaves (Bonne et al., 2015; Gonzalez-Herrero et al., 2022; Gorodetskaya et al.,
2023; Wille et al., 2024a,b), extensive surface melting through foehn warming, cloud radiative
impacts, or rain-on-snow processes (Bozkurt et al. 2018; Gorodetskaya et al. 2023; Zou et al.
2023; Mattingly et al., 2020, 2023), sea ice decline (Zhang et al. 2023; Liang et al. 2023; Francis
et al. 2020; Li et al. 2024), as well as intense snow accumulation over the ice sheets
(Gorodetskaya et al. 2014; Adusumilli et al. 2021; Wille et al. 2024a, b) in both Antarctica and
the Arctic. In regions characterized by intricate topography, like the Antarctic Peninsula, ARs
may strike the mountain range, leading to substantial rainfall and snowfall on the upwind side
owing to orographic lifting (Gorodetskaya et al. 2023). Meanwhile, foehn warming on the
leeward side can intensify surface melting and downslope winds, contributing to ice-shelf and



sea ice weakening and potential disintegration (Bozkurt et al. 2018; Wille et al. 2022; Zou et al. 2023).

In polar areas, ARs can also interact with other weather systems, making an interconnected and
multi-scale impact on extreme weather events. For instance, in March 2022, tropical convection-
induced Rossby wave activities led to an extraordinary AR intrusion into the East Antarctic ice
sheet, with an amplified warm conveyor belt along the coastline due to abundant latent heat
release from additional moisture (Wille et al. 2024a). Consequently, the intensified coupled low-
pressure and blocking high system strengthened the AR's force and its elongated characteristics,
enabling robust moisture penetration deeper inland and exerting a widespread influence on the
ice surface (Wille et al. 2024b). During March 16th–18th, this AR had a maximum vertical-
integrated water vapor transport (IVT) that exceeded 1000 kg m$^{-1}$ s$^{-1}$ offshore before hitting the
East Antarctic coast with over 1000% of normal water vapor transport (measured as time-
integrated IVT, T-IVT) penetrating into the inland area of the East Antarctica (Fig. 1a). The 3-
day T-IVT is approximately 210% of the previous maximum over this region since 1981 (Fig.
1b). In addition to extreme precipitation, this extraordinary AR caused a record-breaking
heatwave. The peak of this heatwave had a temperature soaring 37°C beyond the climatological
average based on ERA5 reanalysis data, and 43°C above the climatological mean temperature in
March as observed at Dome C station (Fig. 1 c and d). The AR brought around 300 Gt of
precipitation over the ice sheet (according to a polar-oriented regional climate model simulation
and two reanalysis datasets), including more than 2 Gt of rainfall along the coast (Wille et al.
2024b). Under a warming climate, the extreme ARs are expected to increase in both frequency
and intensity (e.g., Warner et al. 2015; Warner and Mass 2017; Payne et al. 2020; Shields et al.
2023). These results have implied that the impacts of ARs on polar regions, which are known to
be vulnerable to climate change, could be enhanced in a warmer climate.





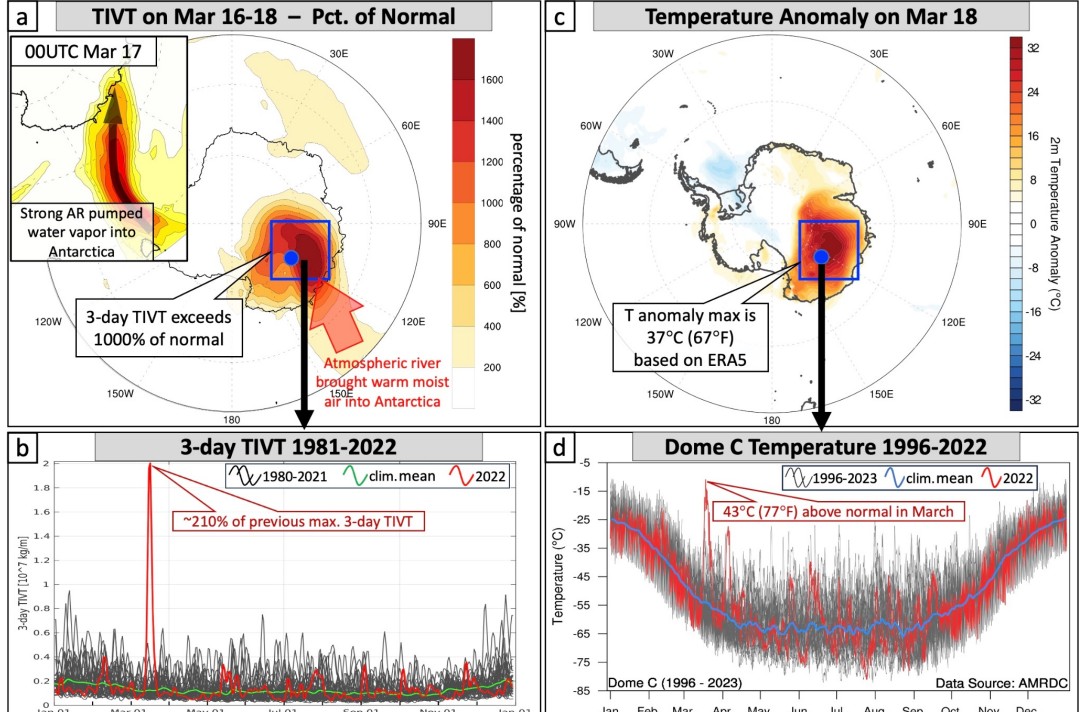

Figure 1. An extreme landfalling AR over East Antarctica during 16–18 March 2022 based on ERA5 reanalysis. (a) Three-day time-integrated IVT (T-IVT) during 16–18 March 2022 as a percentage of normal (mean 3-day T-IVT during 1980-2021); the sub-panel at the top left shows the IVT (colors starts from 200 kg m$^{-1}$ s$^{-1}$ with an increment of 100 kg m$^{-1}$ s$^{-1}$) at 00 UTC on 17 March. (b) Temperature anomaly on 18 March 2022. (c) Time series of averaged 3-day T-IVT within the blue box in panel (a) for 2022 (red), 1980-2021 (black), and climatological mean (1980-2021, green). (d) Time series of 3 hourly observed temperatures at Dome C station (blue dot in panel a and c) for 2022 (red), 1996-2021 and 2023 (gray), and climatological mean (1996-2023 mean, blue).

Recent studies have made important progress in tracking ARs using polar-specific algorithms (Wille et al. 2019, 2021; Gorodetskaya et al 2014, 2020; Viceto et al. 2022; Mattingly et al. 2023) and global algorithms (Guan and Waliser, 2019; Rutz et al., 2019; Guan et al., 2023). These polar AR algorithms identify ARs as objects in space with AR features using flexible thresholds. However, Ralph et al. (2019) suggested that it is also useful to identify ARs from an Eulerian perspective (defining an AR as a sequence of relevant meteorological conditions at a specific location of interest), especially for many practical, on-the-ground applications and





communications. With the increasing interest in ARs in both the Arctic and Antarctica, it is imperative to establish an objective and consistent framework for quantifying their strength and impacts, which is usually determined by both the intensity and duration of the AR event. Ralph et al. (2019) introduced an AR scale (hereafter, Ralph 2019 AR Scale) to characterize their strength and impacts based on the duration of the AR condition (defined as IVT > 250 kg m$^{-1}$ s$^{-1}$)

and the maximum IVT during the AR at a specific location from an Eulerian perspective. The Ralph 2019 AR Scale has been widely used in scientific research, weather forecasts, and media reports. However, this scale was primarily developed based on the climatology of IVT at middle latitudes. The standard scale can miss ARs in polar regions that have a profound effect but nevertheless fail to meet the minimum IVT threshold of 250 kg m$^{-1}$ s$^{-1}$ due to the extremely low

temperature and thus moisture content characterizing polar regions. For example, Figures 2 a-b show a well-defined landfalling AR and the corresponding precipitation at the Ross Ice Shelf on December 3$^{rd}$, 2022; and Figures 2 c-d show a landfalling AR at northeast Greenland with a maximum precipitation rate over 10 mm per 6 hours on November 15$^{th}$, 2021. However, both cases are not identified as ARs at the coast according to the Ralph 2019 AR Scale due to low

IVT: in the first case the duration of time when the IVT exceeds 250 kg m$^{-1}$ s$^{-1}$ is shorter than 24 hours, and in the second case the IVT never meets the minimum threshold for AR condition at the coasts. Obviously, the Ralph 2019 AR Scale developed based on the climatology of IVT at middle latitudes is insufficient for polar regions.

This study aims to introduce an extended AR scale for polar regions based on the Ralph 2019

AR Scale and examine the characteristics of polar ARs by utilizing the extended scale. We first examine the climatology of IVT in polar regions with a focus on the Antarctic and Greenland coastlines. Based on the IVT climatology and many AR cases in polar regions, we introduce an extended version of the AR scale tuned to polar regions. Additional IVT thresholds for defining AR conditions are included to capture low-IVT ARs in both the Arctic and Antarctic. The

extended AR scale is then used to examine AR frequency, seasonality, historical trends, and associated impacts (precipitation and surface melt) in the Antarctic and Greenland coastal regions. Finally, the AR scale forecast products for the Antarctic coast developed by the Center for Western Weather and Water Extremes (CW3E) and its application are introduced.



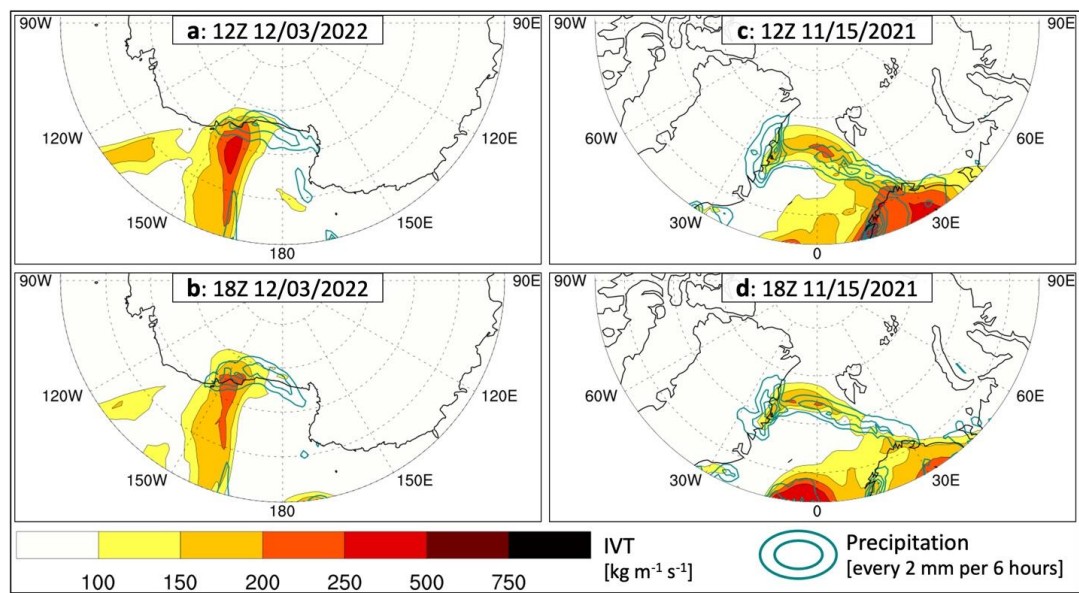


Figure 2. A landfalling AR near the Ross Ice Shelf in Antarctica at 12Z (a) and 18Z (b) on December 3rd, 2022. The colors are IVT (kg m$^{-1}$ s$^{-1}$) and the cyan contours show 6-hour precipitation amount (every 2 mm) based on ERA5. (c) and (d) are the same as (a) and (b) but for a landfalling AR case over the East Greenland coast at 12Z and 18Z on November 15th, 2021.


## 2. Extended AR scale for polar regions

### 2.1 Data

The IVT and precipitation data used in this study are from the fifth-generation reanalysis dataset from the European Centre for Medium-Range Weather Forecasts (ERA5, Hersbach et al., 2020).

The ERA5 data was obtained on a regular latitude-longitude grid with a spatial resolution of 1.0° x 1.0° and a 6-hourly temporal resolution, from January 1979 to December 2022. The ERA5 IVT was vertically integrated from the surface to the atmosphere top in the reanalysis model. The 3-hourly Automatic Weather Station (AWS) observations at Dome C station served as a valuable illustration of the March 2022 heatwave in EA triggered by an AR (Fig. 1d). The dataset,

spanning from 1996 to 2022, has undergone quality control and is archived at the Antarctic Meteorological Research and Data Center (AMRDC).

The daily surface melt data from 1980 to 2020 was retrieved from passive microwave radiometer data of the Scanning Multichannel Microwave Radiometer (SMMR) and the Special Sensor



Microwave/Imager (SSM/I), as documented in previous studies by Torinesi et al. (2003) and
Picard and Fily (2006). The data was obtained at a 25-km spatial resolution and was interpolated
to the same 1.0°x1.0° latitude-longitude grid as the ERA5 data using a nearest-neighbor
interpolation. When the melt data value is 0, it indicates a no melt day; when the value is 1, it
indicates a melt day. The missing data is labeled with -10. Thus, the surface melt data used in
this study only include the melt frequency (melt or not).


## 2.2 Climatology of IVT in polar regions

As a first step in developing an appropriate extended AR scale for polar regions, we examine the
variations in climatological mean IVT between polar regions and low-to-middle latitudes. In
general, the climatological mean IVT, calculated over the investigated period from 1979 to 2022,

is notably lower along the Antarctic and Greenland coasts compared to the IVT at the middle
latitudes (Fig. 3). In the southern hemisphere (Fig. 3a), the mean IVT typically exceeds 100 kg
m$^{-1}$ s$^{-1}$ at the middle latitudes with a maximum (> 250 kg m$^{-1}$ s$^{-1}$) over the central South Atlantic
Ocean and Indian Ocean, respectively. However, there is a rapid decrease in IVT as the latitude
increases (Fig. 3a). Along the Antarctic coast, the mean IVT is around only 25 kg m$^{-1}$ s$^{-1}$ and

falls below 25 kg m$^{-1}$ s$^{-1}$ over most of the interior due to the extremely low temperature and
moisture. Meanwhile, the northern hemisphere exhibits a peak in mean IVT over the North
Atlantic Ocean and another maximum (> 250 kg m$^{-1}$ s$^{-1}$) over the North Pacific Ocean, aligning
with the locations of storm tracks in the two ocean basins (Fig. 3b). Along the Greenland coast,
the mean IVT ranges from 25 to 50 kg m$^{-1}$ s$^{-1}$(Fig. 3b). The substantial difference in IVT

between polar regions and middle latitudes suggests that the IVT minimum threshold based on
mid-latitude IVT in the Ralph 2019 AR Scale is insufficient for polar regions.



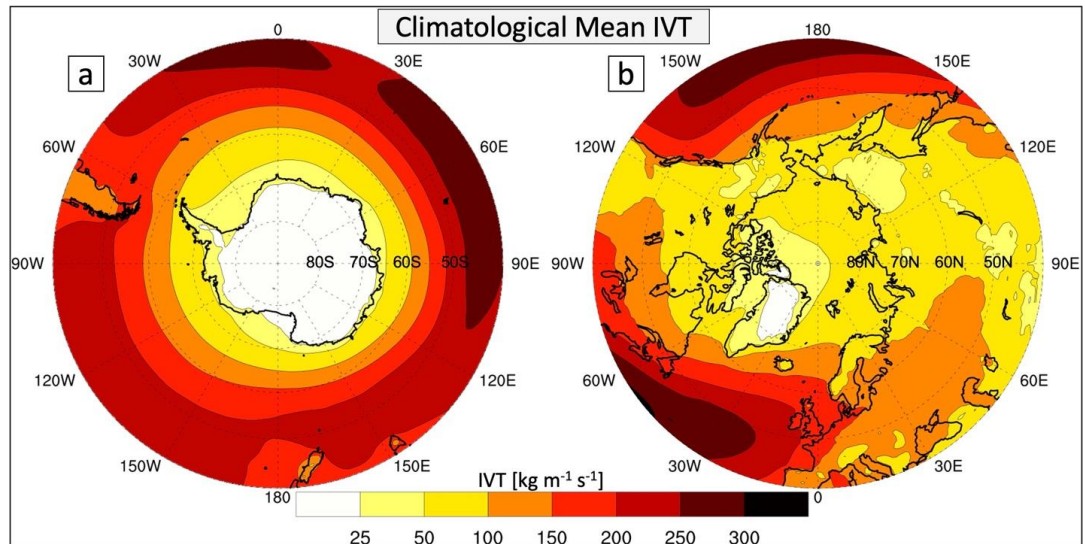

Figure 3. Climatological mean IVT (kg m$^{-1}$ s$^{-1}$) over the Southern Hemisphere (a) and Northern
Hemisphere (b) in 1979–2022 based on the ERA5 reanalysis data.

To further understand the climatology of IVT along the Antarctic and Greenland coasts, the IVT
frequency distribution was calculated (Fig. 4) using 6-hourly IVT values along the Antarctic and
Greenland coastlines (bold black line in Fig. 4 b and c) from 44 years of ERA5 data (1979–2022;
Fig. 4a). For the Antarctic and Greenland coasts, respectively, 77% and 79% of the IVT values
are concentrated in bins lower than 50 kg m$^{-1}$ s$^{-1}$, with only 7.4% and 6.6% of IVT values higher
than 100 kg m$^{-1}$ s$^{-1}$. The percentages of IVT decrease with the thresholds (100, 150, 200, and 250
kg m$^{-1}$ s$^{-1}$) quickly and only 0.3% and 0.4% of IVT exceed 250 kg m$^{-1}$ s$^{-1}$ (subpanel at the top
right of Fig. 4a). The variability of the percentages for each bin (vertical bars) is larger along the
Greenland coast than along the Antarctic coast, indicating that the water vapor transport around
Greenland has a larger variability, which might be related to the variability of the North Atlantic
storm track.

Because this study focuses on polar ARs, the southern part of Greenland with latitude south of
the polar circle (~67°N), was excluded from our analysis, including the calculation of IVT
distribution (Fig. 4c). This narrow part, including 15% of the coastal grids, extends to 60°N and
is surrounded by a relatively warm ocean, so the temperature, moisture, and therefore the IVT
are significantly higher than the coast within the polar circle. Excluding this region has a small



impact on the IVT distribution; the percentage of IVT samples higher than 100 kg m$^{-1}$ s$^{-1}$ would

increase from 6.6% to 8.9% if it were included. Similar analysis and results for the entire

Greenland ice sheet are given in the Supplement (Figs. S1 and S2). It is worth noting that the

northern tip of the Antarctic Peninsula also extends northward beyond the polar circle (~67°S).

However, the coastal grids out of the polar circle are only 3% of the total coastal grids of

Antarctica. Therefore, those grids are not excluded since their impact on the results is negligible.

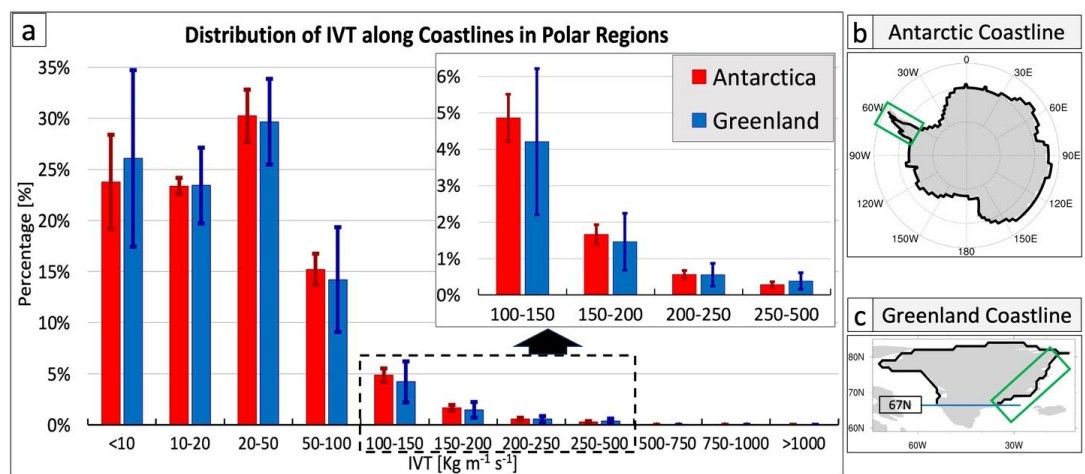


Figure 4. (a) Averaged distribution of IVT values based on 6-hourly samples along the coastlines
of Antarctica and Greenland in polar regions from 1979 to 2022, using ERA5 reanalysis data.
The vertical bars denote the spread (one standard deviation) of the frequency over the 44 years.
Panels (b) and (c) depict the coastlines of Antarctica and Greenland in polar regions (thick black

lines), respectively. The green boxes in (b) and (c) highlight the coastlines of the Antarctic
Peninsula and East Greenland used for the analysis of the AR frequency trend in Figure 13.

## 2.3 Extended AR scale for polar regions

In addition to the analysis of IVT climatology along the Antarctic and Greenland coast, we also

examined many AR cases in the polar regions, such as the two AR cases in Fig. 2. Based on the

IVT climatology and the polar AR case studies, 100 kg m$^{-1}$ s$^{-1}$ is selected as the IVT minimum

threshold to define AR conditions for both polar regions. This new threshold is roughly the 93$^{rd}$

percentile of the IVT values along both the Antarctic and Greenland coasts. In addition, 150 and



200 kg m$^{-1}$ s$^{-1}$ are also selected as the thresholds for the AR scales in polar regions based on the
IVT distribution in Fig. 4.

Therefore, we extend the Ralph 2019 AR Scale to include 3 additional ranks specifically for
polar regions as shown in Fig. 5. The extended minimum IVT thresholds are as follows: 100 kg
m$^{-1}$ s$^{-1}$ for AR Polar 1 (AR P1), 150 kg m$^{-1}$ s$^{-1}$ for AR P2, and 200 kg m$^{-1}$ s$^{-1}$ for AR P3.
Meanwhile, the thresholds for the other ranks (AR1 – AR5) remain the same as the Ralph 2019
AR Scale. Following the Ralph 2019 AR Scale, the extended version of the AR scale is also
defined based on an Eulerian perspective. In other words, the AR scale is defined for a specific
location, so an AR event is a sequence of high-moisture transport conditions, which are closely
associated with variant meteorological conditions, such as the precipitation amount and rate (e.g.,
Ralph et al. 2013; Martin et al. 2018). The AR scale of an event is based on its duration and
maximum intensity of IVT at a given location. For a given location, the duration of an AR
matters since, with the same IVT, when an AR lasts for a longer time, it has a higher impact
(e.g., more precipitation), and vice versa. Following the Ralph 2019 AR Scale, "weak" polar
ARs (maximum IVT > 100 kg m$^{-1}$ s$^{-1}$ and < 150 kg m$^{-1}$ s$^{-1}$, but with a duration < 24 hours) will
not receive a ranking on the polar AR scale, as represented by the gray part in Fig. 5. To
determine the AR scale, there are four steps as described below.

Step 1: Pick a location of interest.

Step 2: Identify the time when IVT exceeds 100 kg m$^{-1}$ s$^{-1}$ at that location. The period when IVT
       continuously exceeds 100 kg m$^{-1}$ s$^{-1}$ is defined as the duration of an AR event.

Step 3: Identify the maximum IVT during the AR event at that location, and then use the chart
(Y axis) in Fig. 5 to initially assign a preliminary AR scale based on maximum IVT.

Step 4: Adjust the rank based on the duration to determine the final AR scale: (1) if the AR
       duration exceeds 48 hours, promote the scale by 1 rank; (2) if the AR duration is less than
       24 hours, demote the scale by 1 rank.




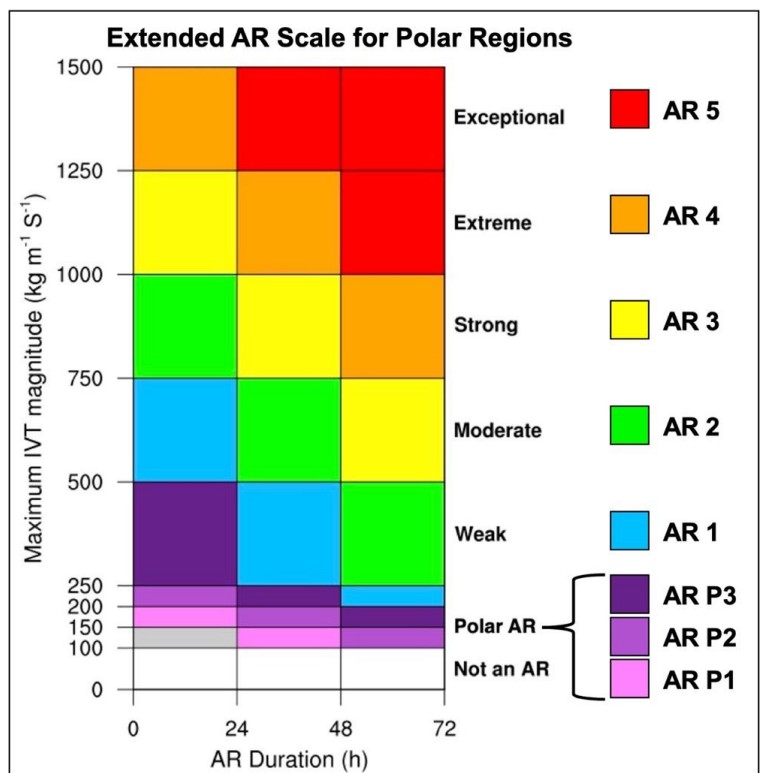


Figure 5. An extended AR scale for polar regions that categorizes AR events based on the duration of AR conditions (IVT ≥ 100 kg m$^{-1}$ s$^{-1}$) and the maximum IVT in the duration at a specific location. This scale includes ranks (AR P1, AR P2, and AR P3) designed specifically for ARs in polar regions.


In Fig. 6, the extreme landfalling AR over East Antarctica in March 2022 (as discussed in Section 1 and shown in Fig. 1) is used as an illustrative example for determining the AR scale in the polar regions. Figures 6 a–c show three snapshots of IVT for this case at 12 UTC on the 16$^{th}$, 17$^{th}$, and 18$^{th}$ of March. The landfall of this long-lasting AR occurred mainly along the coast
between 110°E – 130°E, with a peak IVT exceeding 800 kg m$^{-1}$ s$^{-1}$. To determine the AR scale, we begin by selecting a location, such as a dot at 125°E on the coastline (dot with a black arrow in Fig. 6d). From the time series of IVT at that location, identify the time with AR conditions (IVT > 100 kg m$^{-1}$ s$^{-1}$), which is 124 hours (depicted in orange part in Fig. 6e). Subsequently, find the maximum IVT during this AR event at that location (845 kg m$^{-1}$ s$^{-1}$), which indicates
that the preliminary AR scale is AR3 according to the chart in Fig. 5. Finally, promote the AR



scale by 1 rank since the duration exceeds 48 hours. Therefore, the final AR scale of this event at 125°E on the coastline is determined to be AR4. Figure 6d shows an example of the maximum AR scale during the week of March 14th – 20th along the Antarctic coast with small dots denoting locations along the coastline. Different colors of the dots along the coast represent various AR

scales or the absence of an AR (white) at the respective locations.

The extended AR scale for polar regions proves effective in capturing even relatively weak AR events. For example, the Antarctic AR case (Figs. 2a, b) is classified as AR P3, and the Greenland AR case (Figs. 2c, d) is identified as AR P2, employing the same procedure as described above.


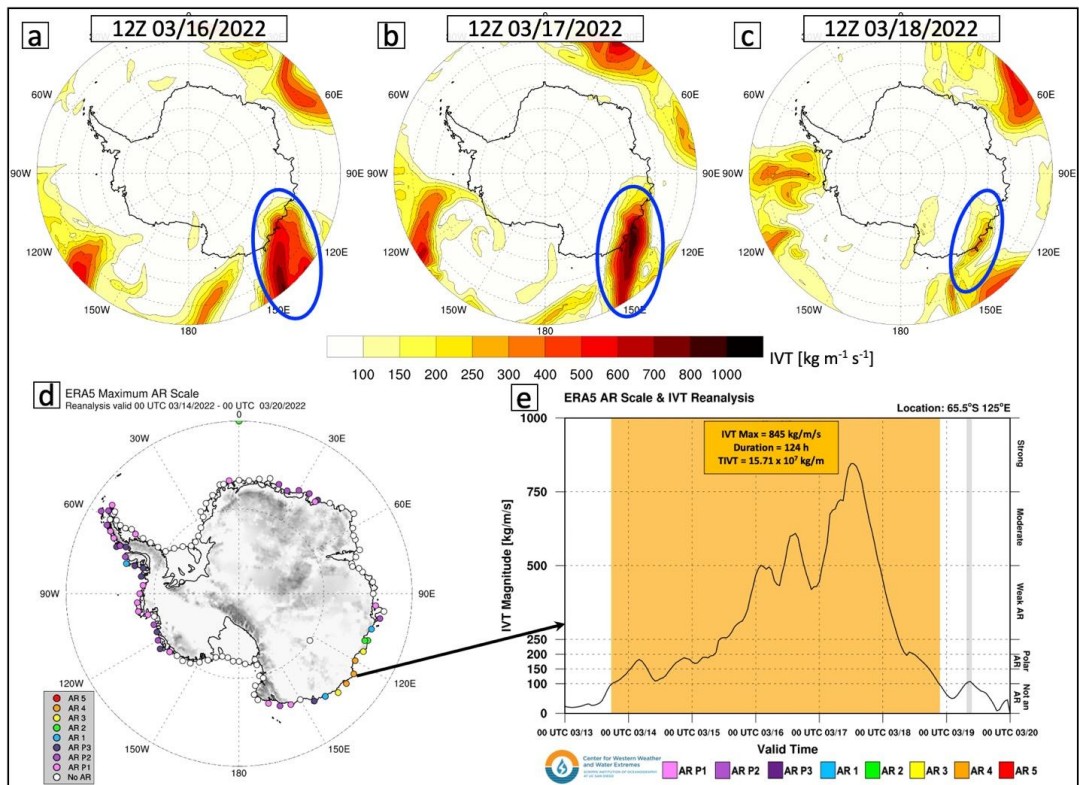

Figure 6. Panels (a)-(c): three snapshots of IVT associated with an extremely strong landfalling AR event (highlighted in the blue ovals) over East Antarctica in the middle of March 2022. The colors in (a)-(c) are the IVT (kg m$^{-1}$ s$^{-1}$) from the ERA5 reanalysis data at 1200 UTC on the 16th,

17th, and 18th of March 2022, respectively. Panel (d): maximum AR scale along the coastline of



Antarctica during the week of 14th–20th March 2022. (e) shows the time series of IVT at the AR landfalling location (65.5°S, 125°E) and the corresponding AR scale.

# 3. Climatology of polar ARs

## 3.1 Frequency of polar ARs

Using the extended AR scale for polar regions, we examine the climatology of the polar ARs along the coastlines of Antarctica and Greenland. The frequency of AR P1 is around 4 events per year per location along most of the East Antarctic coast and the frequency decreases rapidly towards the inland area (Fig. 7a), which is mainly due to the extremely low temperature and thus

low moisture over the interior of Antarctica. The frequency of AR P1 is slightly higher over the West Antarctic coast, and there are more AR P1 events over the inland area close to the coast compared to a similar area in East Antarctica (Fig. 7a). It is easier for the warm and moist air to penetrate the inland area over West Antarctica as the elevation there is relatively low (e.g., over the Ross Ice Shelf; Nicolas and Bromwich, 2011). The frequency of AR P1 at the Antarctic

Peninsula (5-10 ARs per year per location) is the highest along the Antarctic coast since the peninsula extends into the ocean at a relatively lower latitude with more moisture and propagating ARs. The frequency of ARs decreases with higher AR scales (Fig. 7). There are only 1-2 AR1 events (Fig. 7d) and fewer than 1 AR2 event (Fig. 7e) per year per location over most of the Antarctic coast. There are few AR4 or AR5 over the Antarctica continent as the IVT

thresholds are so high that only a few cases were ranked as AR4 during the last few decades based on ERA5.



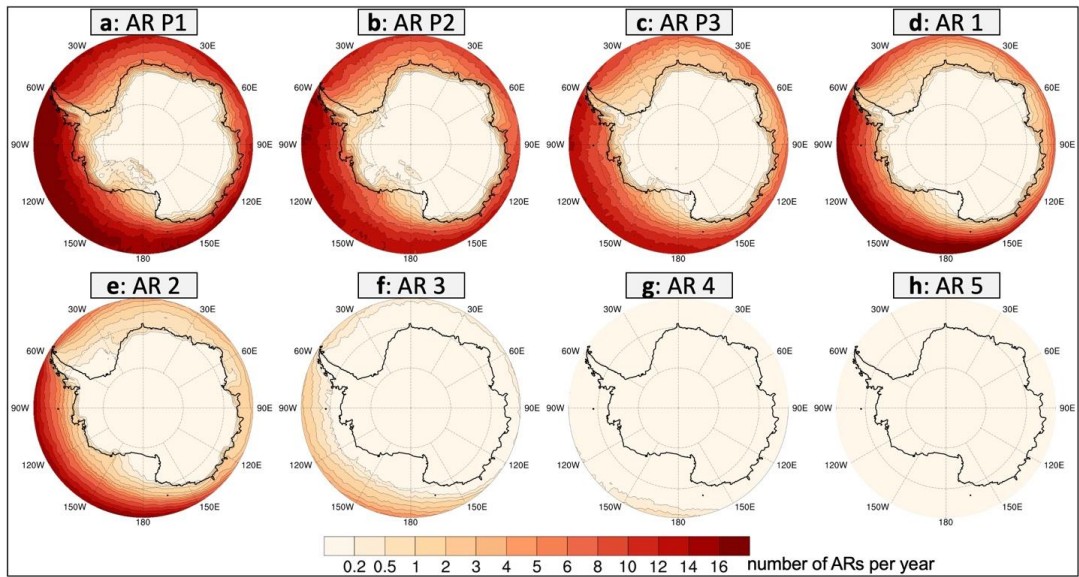

Figure 7. Averaged AR frequency (number of events per year per location) over the South Polar
Region in 1979–2022 based on ERA5 according to the new polar AR scale: (a) AR P1, (b) AR
P2, (c) AR P3, (d) AR 1, (e) AR 2, (f) AR 3, (g) AR 4, and (h) AR 5.

The AR scale frequency over the Arctic (Fig. 8) is generally higher compared to Antarctica. ARs
intruding into the Arctic mainly originate from the North Atlantic Ocean, making their way

through the gap between Greenland and Northern Europe. As a result, the AR frequency reaches
its peak within that gap, amounting to approximately 12 ARs per year per location for AR P1 and
P2, around 10 ARs for AR P3, and approximately 12 for AR1 and AR2 within the polar circle. A
second gap with AR intrusions is over the Bering Strait, where ARs originate from the North
Pacific Ocean. However, the AR frequency is considerably lower over that narrow strait

compared to the North Atlantic gap. The relatively strong ARs (AR3, AR4, and AR5) are mainly
over the ocean, and the frequency decreases rapidly with the higher AR scales, similar to the
Antarctic.

In this study, for the Arctic region we focus on the ARs over Greenland as the ice sheet there and
its melt is a critical component of global climate change (Alley et al., 2005; Dutton et al., 2015).

Similar to Antarctica, the AR frequency decreases rapidly from the Greenland coastal area
towards the interior. The AR frequency maximum along the Greenland coast is situated from
southern Greenland to the southeast coast, characterized by not only relatively higher



temperature and moisture but also enhanced AR activities associated with the North Atlantic storm track.


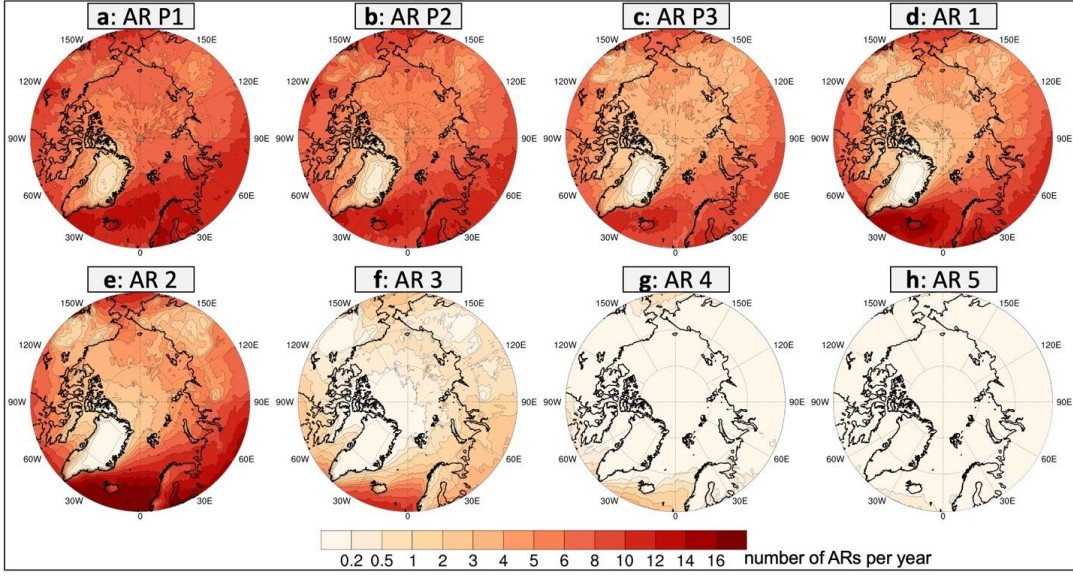

Figure 8. Same as Fig. 7 but for the North Polar Region.

As described above, the ARs are mainly concentrated along the coastal areas of Antarctica and

Greenland; thus the AR frequency along these coastlines are further analyzed. Figure 9 shows the distribution of AR scale frequency along the Antarctic and Greenland coastlines. For the Antarctic coast, there are 5.501 AR P1 events per year per location, and the number decreases rapidly from AR P1 to AR5. There are only 1.747 and 0.776 ARs per year per location classified as AR1 and AR2, respectively. The average annual number is only 0.011 events for AR3 and

0.001 for AR4, which suggests that AR3 or stronger AR events are rare along the Antarctic coast, although they are common at the middle latitudes, like the U.S. West Coast. Most of these relatively strong ARs are located over the Antarctic Peninsula (Fig. 7). The AR4 landfalling event over East Antarctica in March 2022 is a record-breaking case, as described in Section 1. In summary, an average of 14.9 ARs per year per location is identified along the Antarctic

coastline.





The distribution of AR frequency along the Greenland coastline is similar to the frequency along the Antarctic coastline, with a rapid decrease from AR P1 to AR5. Although the average annual number of ARs along the Greenland coast (12.1 ARs per year per location) is lower than that along the Antarctic coast, the frequency of relatively stronger ARs (AR2 – AR4) is higher along the Greenland coast (Fig. 9b). The AR frequency along the entire Greenland coast (bold black line in Fig. S1c) is also calculated. The average annual number of ARs increases to 17.0 per year per location when including the coast of the southern Greenland part (latitude > 67°N). This increase is attributed to the significantly stronger IVT around the coast outside of the polar circle.

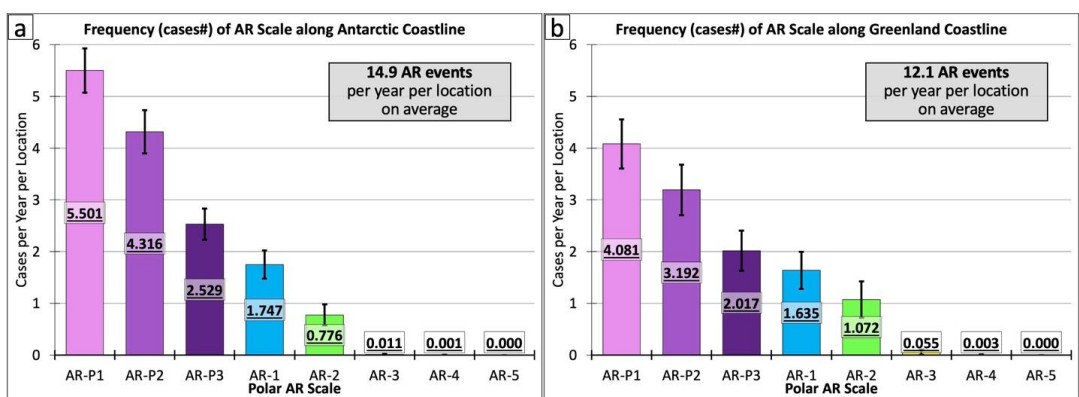

Figure 9. Averaged frequency of ARs along the coastline of (a) Antarctica and (b) Greenland according to the new polar AR scale based on the ERA5 reanalysis data from 1979 to 2022. These coastlines are highlighted by bold black lines in Figs. 4 b and c.

## 3.2 Seasonality of polar ARs

To understand the seasonal variations in AR frequency in the polar regions, a detailed analysis of the seasonality of AR occurrence along the Antarctic and Greenland coastlines is conducted (Fig. 10). Overall, the AR frequency over the Antarctic coast (Fig. 10a) is low in the cold season with a minimum in August (3.2% of the time under AR conditions), and high in the warm season with a maximum in January (8.9% of the time under AR conditions). The relatively weaker ARs (e.g., AR P1 and AR P2) have a weak seasonality while the stronger ARs (e.g., AR1 and AR2) have a stronger seasonality. For example, AR P1 frequency has a maximum of 1.6% in February and a minimum of 0.9% in September; meanwhile, AR2 has the same maximum of 1.6% in January





but a minimum of only 0.2% in August. Along the Greenland coast, the AR frequency is also

relatively high in the warm season with a maximum of 16.0% in July (Fig. 10b). In contrast to
the Antarctic coastline, the Greenland coast has a strong seasonality for all AR scales. ARs along
the Greenland coast primarily occur during the boreal summer, likely influenced by increased
moisture availability during this season and the poleward shift of the North Atlantic storm track
in summer.

The seasonality of AR frequency over the Antarctic coast in this study is different from the
findings of Wille et al. (2021). Their study indicated a greater AR frequency in June, July, and
August (JJA) over Antarctica. This disparity is not unexpected, as Wille et al. (2021) defined
ARs using the meridional component of IVT and a relatively high threshold of IVT (the 98th
percentile), whereas our study employs a fixed minimum threshold (100 kg m$^{-1}$ s$^{-1}$) for the total

IVT. Since the meridional IVT is greatly impacted by extratropical cyclones, the seasonality of
AR frequency in Wille et al. (2021) is closely related to the occurrence of extratropical cyclones,
which are more active during JJA in the Southern Hemisphere.

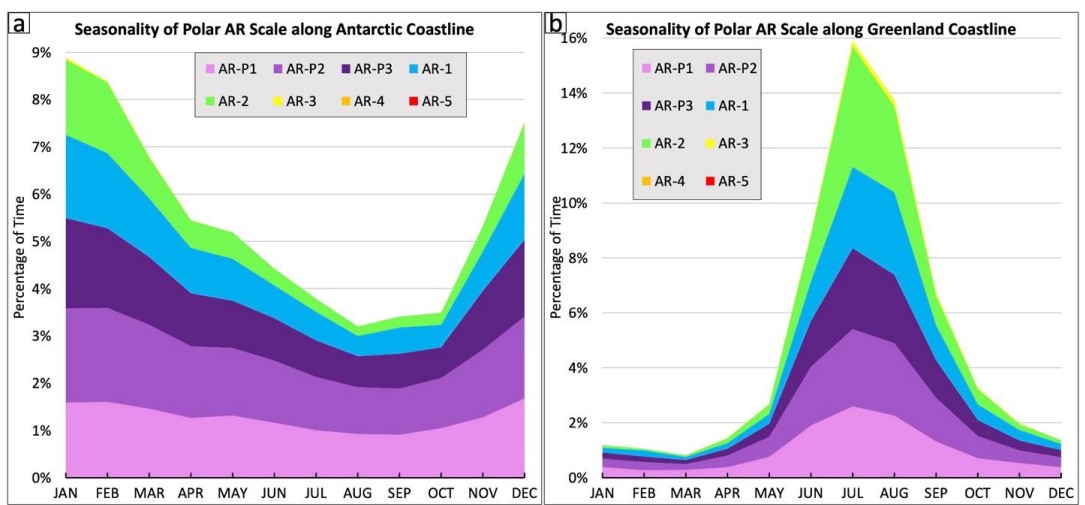

Figure 10. Seasonality of AR frequency along the coastline of Antarctica (a) and Greenland (b)
according to the new polar AR scale in 1979–2022 based on the ERA5 reanalysis data. Different
colors denote the ARs at different scales.

## 3.3 Interannual variability of polar ARs



This section focuses on the interannual variabilities of AR frequency along the Antarctic and
        Greenland coasts. Along the entire Antarctic coast, the interannual variability is minimal with no
        significant trend observed in the frequencies of all ARs (Fig.11a). While the AR frequency along
        Greenland exhibits a slightly increasing trend, it is not statistically significant (Fig.11b). As the
        AR frequency has large regional variability, the interannual variabilities of AR frequency in

specific subdomains of Antarctica and Greenland are also examined. The AR frequency along
        the coast of the Antarctic Peninsula (green box in Fig.4b) has a larger interannual variability and
        a statistically significant increasing trend (+0.89 ARs per decade, 90% confidence interval),
        especially during the recent decade (Fig.11c). This aligns with findings from previous studies,
        which show that there is an increasing trend in AR frequency over West Antarctica (e.g.,

Maclennan et al. 2023). This increasing trend may be related to the poleward shift of
        extratropical cyclones (Chang et al., 2012; Yin 2005). Along with the location and strength of the
        Amundsen Sea Low, cyclone activities significantly affect regional circulation, thereby
        impacting AR characteristics in West Antarctica and the Antarctic Peninsula (Coggins and
        McDonald, 2015; Wille et al., 2021). It also might be related to the decrease of sea ice and thus

increased open water offshore, which can enhance the availability and delivery of water vapor in
        ARs (Kromer and Trusel, 2023).

        The increasing trend of AR frequency is also found along the Greenland coast (Fig. 11b), with
        most of the increase attributed to relatively stronger ARs (AR P2 or stronger ARs). Along the
        coast of East Greenland (green box in Fig.4c), it exhibits a statistically significant increasing

trend (+1.21 ARs per decade, 95% confidence interval), and most of the increase in AR
        frequency comes from relatively stronger ARs (Fig.11d). Similar to the increase trend in the
        Antarctic Peninsula, the increasing trend in the recent decade is the strongest along the coast of
        East Greenland.



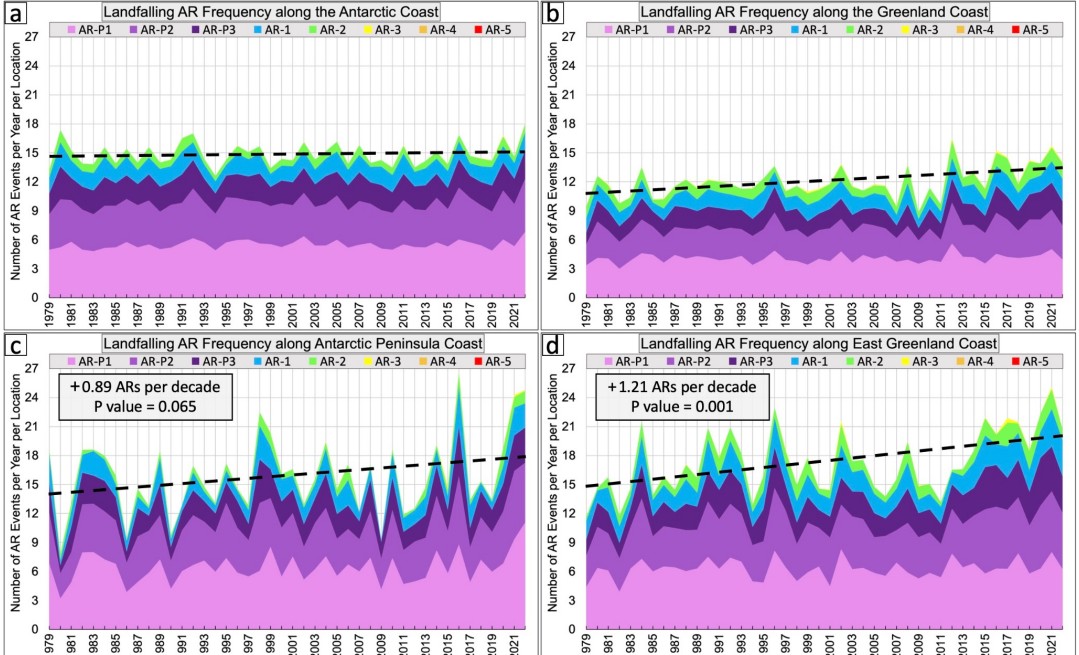

Figure 11. (a) The time series of averaged frequency of landfalling ARs along the Antarctic coastline according to the new polar AR scale in 1979–2022 based on ERA5. Different colors denote the ARs at different scales and the dashed black line is the linear trend for the frequency of all ARs. (b), (c), and (d) are the same as (a) but for the coastline of Greenland in the polar region, Antarctic Peninsula (highlighted in the green box in Fig. 4b), and East Greenland (highlighted in the green box in Fig. 4c) respectively. The P values in (c) and (d) are calculated from a T-test.

## 4. Impacts of polar ARs

### 4.1 Precipitation and polar ARs

ARs are always closely associated with precipitation. Recent studies have explored the impacts of ARs on precipitation in polar regions (Wille et al., 2021; Maclennan et al., 2022; Box et al., 2023). In this section, the contributions of ARs to the annual precipitation along the Antarctic and Greenland coasts are investigated and categorized based on the AR scale. At a given location, precipitation that occurs during an AR event and 12 hours before or after the AR event is classified as AR-related precipitation.



Along the Antarctic coast, ARs contribute 32.4% of annual precipitation on average based on ERA5 (Fig.12a). On average, AR P1 and P2 events each contribute approximately 8.5% of the annual total precipitation amount, and their contributions decrease with higher AR scales due to

their lower frequency of occurrence. AR3 and AR4 events contribute less than 0.1% of the annual precipitation on average due to their substantially low occurrence frequency. However, if a relatively strong AR (e.g., AR4) event does occur, it usually has a remarkable impact on the precipitation amount. For instance, an AR4 event at a given location along the Antarctic coast can contribute 8.6% of the annual precipitation there on average (Fig.12c). The contribution of

each AR event is usually proportional to the AR scale, and it decreases from 4.6% for an AR3 to 1.5% for an AR P1 (Fig. 12c). There is no data for AR5 along the Antarctic coast because no AR5 case has been identified along the Antarctica coast since 1979 based on ERA5.

Along the Greenland coast, the total contribution of ARs to the annual precipitation is 31.8% (Fig.12b), which is close to the result for Antarctica (32.4%). However, the contributions to

annual precipitation from AR P1 – AR2 are comparable, ranging from 5.7% to 7.0%. That is because the mean contribution of each AR event increases quickly from 1.7% for AR P1 to 6.4% for AR2 (Fig.12d). This increase compensates for the decrease in AR frequency. While the contribution of each AR event increases to 7.2% and 7.6% for AR3 and AR4, the frequencies of these two scales are below 0.1 event per year per location, so their total contribution to the

annual precipitation amount is below 1.0%.

Overall, weak and moderate ARs (AR P1 – AR2) are responsible for the majority of the AR-related precipitation amount along the Antarctic and Greenland coasts due to their relatively high frequency. Meanwhile, the strong and extreme ARs (e.g., AR3 and AR4) are usually associated with extreme precipitation in these regions, although their frequency is relatively low.






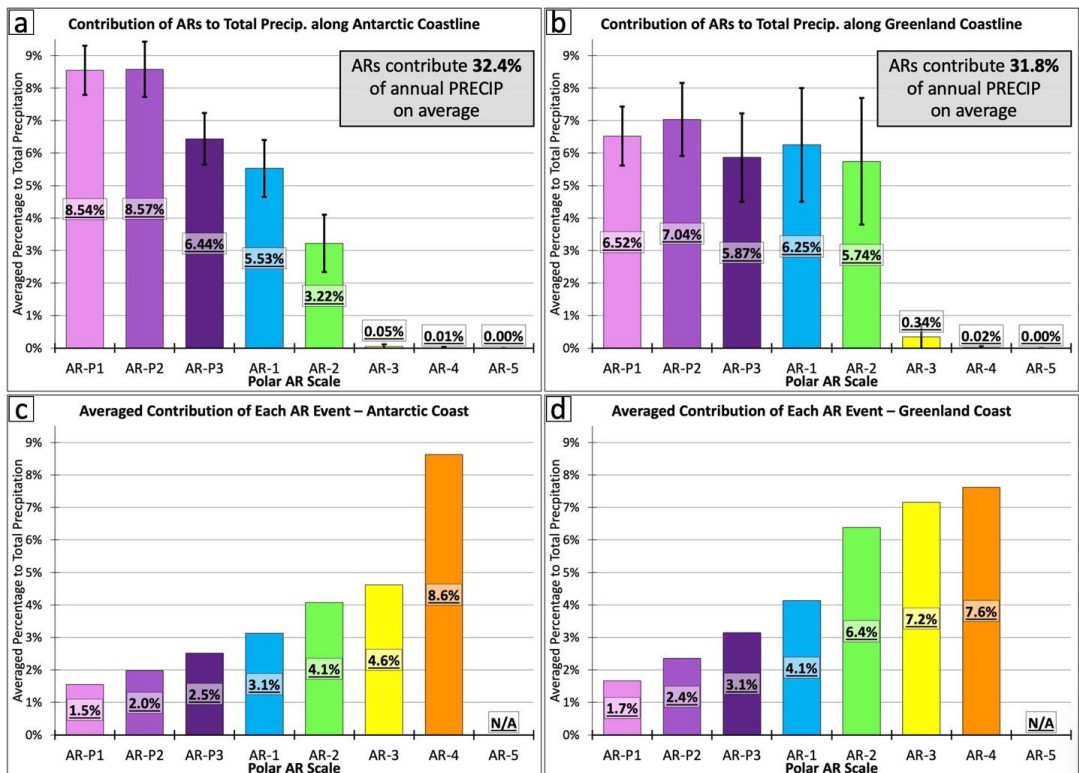

Figure 12. Averaged contribution of ARs to annual precipitation amount along the coastlines of Antarctica (a) and Greenland (b) in 1979–2022 based on the ERA5 precipitation. (c)-(d) are the same as (a)-(b) but for the averaged contribution of each AR event in these regions.


## 4.2 Surface melt and polar ARs

In addition to precipitation, ARs are also related to surface melt in polar regions (e.g., Mattingly et al., 2018; Wille et al., 2022). Surface melting in polar regions is highly correlated to near surface air temperature (Trusel et al., 2015). ARs can potentially trigger widespread melting or

intense snow accumulation, depending on temperature conditions. This section explores the contribution of ARs to surface melt along the Antarctic and Greenland coasts based on the AR scale. Since the melt data is daily, we first identify the AR days at a given location. These AR days are defined as the ones exhibiting AR conditions at any time points (00, 06, 12, or 18 UTC) at the given location. Mattingly et al. (2023) found that there is a delay of 18–24 hours between

AR landfall and the associated surface melt in Greenland, and the time lag is even longer for





strong ARs. Therefore, if the melt occurs on the identified AR days or within one day before or two days after the AR days, it is classified as AR-related surface-melt days.

For the Antarctic coast, Fig. 13a shows the mean number of days with surface melt in austral summer (DJF) from 1980 to 2020. The AR-related surface-melt days are categorized based on the extended AR scale (different colors) and the melt days not associated with ARs are labeled by gray. Both the numbers of total melt days and AR-related melt days along the Antarctic coast have a slight decreasing trend but are not statistically significant. Over the Antarctic Peninsula and West Antarctica, where surface melting is most prevalent, there has been a cooling trend since the 1990s and 2000s (Jones et al., 2019; Zhang et al., 2023), which may contribute to the slight decreasing trend of melt days. On average, ARs contribute to 37% of the surface-melt days with a relatively large interannual variability (Fig.13c). The percentage of AR-related melt days to total melt days ranges from 23% to 52%.

Along the Greenland coast, there is also a decreasing trend in the numbers of total melt days and AR-related melt days (statistically significant at 90% confidence level) in boreal summer (JJA). This trend is mainly attributed to the large numbers of melt days during the 1990s (Fig.13b). The contribution of ARs to the total melt days is 31% and exhibits a large interannual variability along the Greenland coast (Fig.13d). The percentage of AR-related melt days to total melt days varies from 11% to 58%.

Overall, ARs are associated with approximately one-third of the surface-melt days in summer along the Antarctic and Greenland coasts. There are other factors (e.g., surface temperature) that have large impacts on surface melting. More studies are needed to better understand the details of trends of the melt days and the contribution of ARs.



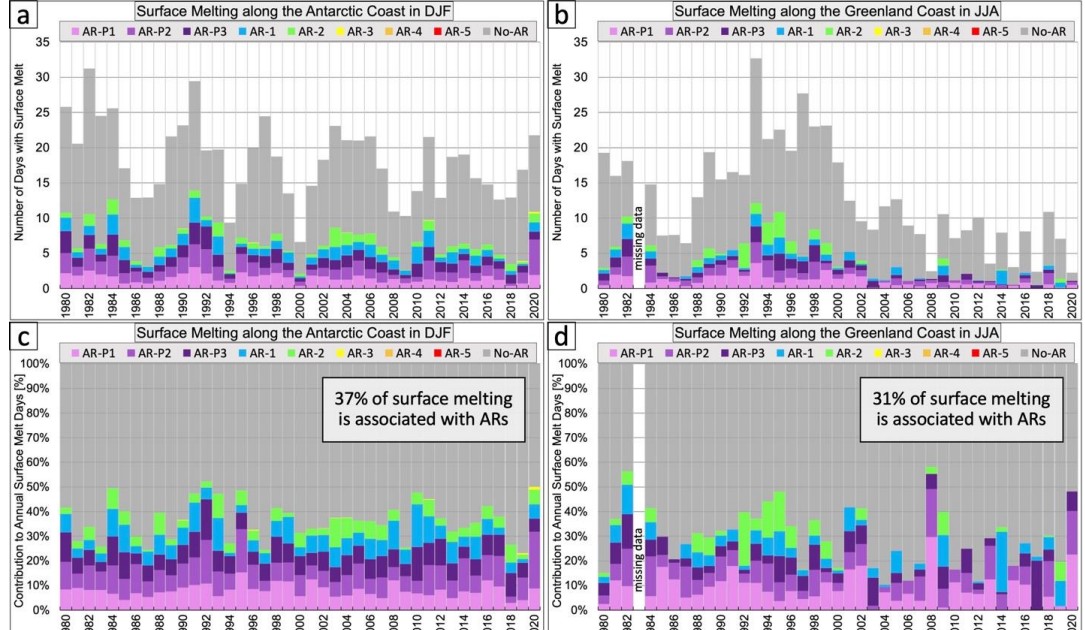

Figure 13. (a) Stacked number of days with surface melting along the Antarctic coast in DJF during 1980-2020; gray indicates surface melting not associated with ARs and the other colors indicate surface melting associated with different AR scales. (b) Same as (a) but for the coast of Greenland and in JJA. (c) Same as (a) but Y-axis is the percentage of surface melting days not associated with ARs (gray) and associated with different AR scales (other colors). (d) Same as (c) but for the coast of Greenland and in JJA.

## 5. CW3E Antarctic AR forecast products

The extended AR scale for polar regions serves as an objective framework to quantify the strength and impact of polar ARs for both scientific research and practical application. It has already been utilized in the forecast products for Antarctica.

Following a highly successful summer campaign, the Year of Polar Prediction in the Southern Hemisphere (YOPP-SH) project initiated targeted observing periods (TOPs), aiming to enhance forecasting skills during non-summer months. The polar AR scale was developed collaboratively during the YOPP-SH Winter TOPs, featuring cooperation between the Center for Western Weather and Water Extremes (CW3E) of Scripps Institution of Oceanography and the Byrd Polar and Climate Research Center at The Ohio State University. Existing AR Scale forecast



tools were then adapted to the new scale using the Global Ensemble Forecast System (GEFS) to
display forecasts of the newly developed Polar AR scale along the Antarctic coast
(https://cw3e.ucsd.edu/arscale_antarctica/). This suite of tools demonstrated reliability in guiding
radiosonde launches from 24 stations during TOPs. Moreover, this scale has proven valuable for
research on AR-associated extreme weather events (Bromwich et al., 2024), as well as their
impacts on the Antarctic ice surface (Wille et al. 2024a, b; Gorodetskaya et al. 2023; Zou et al.
2023).

The AR Scale ensemble diagnostics tool displays the forecasted timing and probability of ARs
making landfall at a given point and the associated Polar AR scale ranking for the next seven
days (Fig. 14). This product is available for all locations shown in the map in Fig. 14a, colored
dots represent the maximum Polar AR scale forecasted over the next seven days with the
enlarged dot representative of the selected location for the other panels. For the selected location
a seven-day forecast of IVT magnitude from each ensemble member as well as the ensemble
mean and +/- 1 standard deviation is displayed in the top left, along with color shading
representing the Polar AR scale based on the ensemble control member (Fig. 14b). The
probability of all Polar AR scale ranking as a function of lead time is displayed in the bottom
left, based on the number of ensemble members forecasting a given ranking at a given lead time
(Fig. 14c). Lastly the timing and magnitude of the Polar AR scale is displayed for each ensemble
member in the bottom right, with text within each colored bar representing the timing and
magnitude of maximum IVT (Fig. 14d). Such detailed insights enable improved situational
awareness, contributing to timely preparedness and effective decision-making for high-impact
events over polar regions.

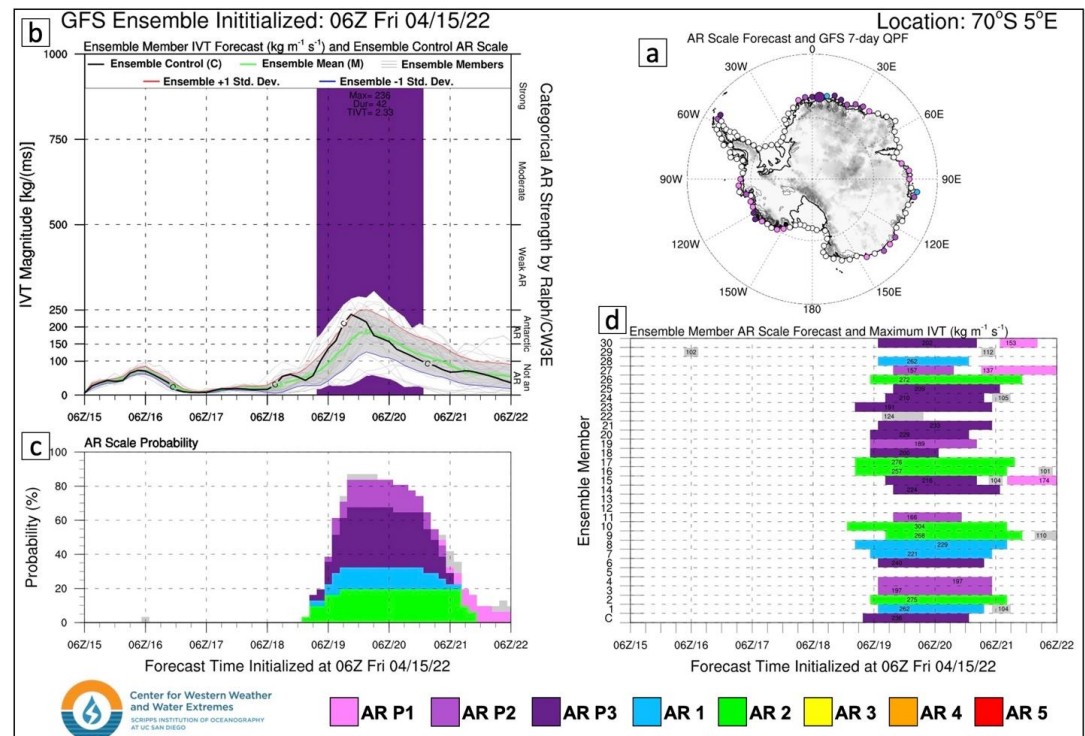

Figure 14. CW3E AR scale ensemble diagnostic forecast tool for 70°S, 5°E from the GEFS. initialized at 06Z 04/15/2022. Dots along the Antarctic coast indicate locations where information such as that shown in other panels can be provided; here other panels refer to the larger point at 70°S, 5°E).(a) Maximum Polar AR scale forecast over the next seven days along the Antarctic Coast colored dots), enlarged dot represents the location shown in panels b-d. (b) Seven day forecast of IVT from each ensemble member (thin gray lines), the ensemble mean (green line), control member (black line) and +/- 1 standard deviation from the ensemble mean (red and blue lines and gray shading). Color shading represents the timing of the Polar AR scale from the control member. (c) Forecasted probability of each Polar AR scale ranking as a function of lead time based on the number of ensemble members predicting an AR. (d) Forecasted Polar AR scale timing and ranking from each ensemble member, text values represent the maximum IVT magnitude and timing during a forecasted AR.

# 6. Conclusions and discussion

Following the Atmospheric River (AR) scale developed by Ralph et al. (2019, Ralph 2019 AR Scale), this study introduces an extended AR scale for the polar regions with a focus on the Antarctic and Greenland coasts. The Ralph 2019 AR Scale was developed based on the IVT





climatology at the middle latitudes, and is insufficient for the polar regions, where the temperature and moisture are extremely low. Based on the climatology of IVT in polar regions,

this study introduces an extended AR scale. This updated scale includes three low IVT minimum thresholds, corresponding to three new ranks, specifically tailored to ARs affecting the polar regions. The scale of an AR event is determined based on its duration (the period when IVT exceeds 100 kg m$^{-1}$ s$^{-1}$) and intensity (maximum IVT) at a specific location. Using the extended AR scale, this study investigates the climatology of AR events in polar regions and categorizes

them based on their strength. In addition, the impacts of ARs on the precipitation and surface melt are explored. Finally, an AR scale forecast tool developed by CW3E is introduced as an example of the application of the new extended AR scale for polar regions.

Unlike many previous AR detection methods (Wille et al., 2019, 2021; Shields et al., 2018; O'Brien et al., 2020), the extended AR scale framework in this study is designed to objectively

quantify the strength and impact of individual AR events from an Eulerian perspective, for a specific location. The extended AR scale provides an objective and concise description of the strength of AR events at the locations of interest, aiming to enhance communications across observation, research, and forecasts for polar regions.

This study explores the impacts of ARs on precipitation and surface melt along the Antarctic and

Greenland coasts based on the extended AR scale. ARs contribute over 30% of the annual precipitation amount along the coasts on average, and the relatively strong ARs are usually associated with extreme precipitation events. During summer, ARs are related to 37% and 31% of the surface melt over the Antarctic and Greenland coasts, respectively. More research is needed to further investigate the details regarding the impacts of ARs in polar regions, like the

AR's contribution to rainfall and snow in polar regions, the mechanism underlying the impacts of ARs on surface melt, and the interactions between ARs and other weather systems. Many previous studies investigated those topics (Wille et al. 2019, 2021; Gorodetskaya et al., 2023; Zou et al., 2023; Zhang et al., 2023; Mattingly et al., 2018, 2023), but including an objective description of AR strength (the extended AR scale) can improve the understanding of ARs'

impacts on polar regions.



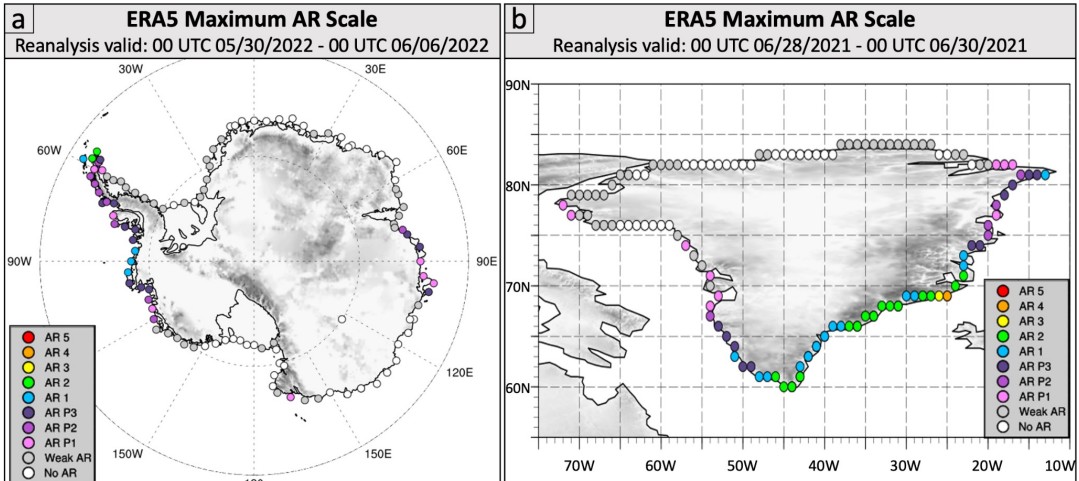

Figure 15: (a) Maximum Polar AR scale identified along the Antarctic coast during 00Z
05/30/2022 – 00Z 06/06/2022 based on ERA5. (b) Maximum Polar AR scale identified along the
Greenland coast during 00Z 06/28/2021 – 00Z 06/30/2021 based on ERA5.

As described in Section 5, the extended AR scale for polar regions has already been used in the

CW3E Polar AR scale forecast tools, which was intensively used in the YOPP-SH Winter TOPs.

The extended AR scale successfully identifies polar AR conditions that would have been missed

by the standard AR scale (e.g., AR P1 – AR P3 in Fig. 15). For instance, a landfalling AR was

ranked as AR P3 over the East Antarctic coast (Fig. 15a) and passed over the region around the

Davis Station on 3–4 June 2022 during the YOPP-SH Winter TOPs (Bromwich et al., 2024).

Another illustrating example is an extreme AR influencing Greenland during 28-30 June 2021

(Fig. 15b). In addition to the main landfalling area ranked as AR1 – AR4, the extended AR scale

captures the AR traveling from the southwest to the east of Greenland over the three days. The

main goal of this extended AR scale for polar regions is to enhance communications across

observation, research, and forecasts for polar regions. The extended scale, along with the newly

developed forecast tool at CW3E, has great potential to enhance situational awareness,

contributing to timely preparedness and effective decision-making for high-impact events over

the polar regions that are acknowledged to be vulnerable to a changing climate.



**Data availability.** The ERA5 Reanalysis data can be found on the Climate Data Store website
from the Copernicus Climate Change Service, https://cds.climate.copernicus.eu/cdsapp#!/home.
The 3-hourly Automatic Weather Station observations at Dome C station can be found on the
AMRDC Data Repository website, https://amrdcdata.ssec.wisc.edu/organization/amrdc. The
daily surface melt data retrieved from SMMR and SSM/I was downloaded from Dr. Ghislain
Picard's website https://snow.univ-grenoble-alpes.fr/melting/.


**Author contributions.** ZZ conducted the analysis of AR scale, IVT, precipitation, and surface
melt. XZ analyzed the temperature data for the extreme AR case in March 2022. MZ performed
the analysis for the interannual variability and historical trend of ARs. BK developed the AR
scale forecast tool. IVG and PMR contributed to the result interpretation for polar regions. FMR
and DHB contributed to the conceptualization of the polar AR scale. ZZ prepared the manuscript
with contributions from all co-authors.

**Competing interests.** The authors declare that they have no conflict of interest.

**Acknowledgements.** ARs play an important role in the surface ice melt in Antarctica and
Greenland, leading to sea-level rise in the highly populated California coastal region. This study
extends the original AR scale to polar regions and aims to enhance communications across
observation, research, and forecasts, which is favorable for California to address the risk
associated with sea-level rise under climate change. We would like to acknowledge the
California Department of Water Resources Atmospheric Rivers Program Phase III for support
for the development of the original AR Scale and associated products (contract 4600014294).
This paper is a contribution to the Year of Polar Prediction (YOPP) international initiative and to
the SCAR scientific research program AntClimNow. I.V.G. thanks the support by the strategic
funding to CIIMAR (UIDB/04423/2020, UIDP/04423/2020), 2021.03140.CERCIND, projects
ATLACE (CIRCNA/CAC/0273/2019) and MAPS (2022.09201.PTDC) through national funds
provided by FCT – Fundação para a Ciência e a Tecnologia. P.R. and X.Z. are grateful to the
National Science Foundation (NSF) for support under awards 2127632 and 2229392. DHB is
supported by NSF grant 2205398. Contribution 1629 of Byrd Polar and Climate Research
Center. We are thankful for the AWS station observations collected at Dome C by the Antarctic
Meteorological Research and Data Center (AMRDC). We extend our thanks to Dr. Ghislain
Picard for sharing the daily surface melt data retrieved from SMMR and SSM/I.



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
