# Peer review of "Figure S1. Same as Figure 4 but the Greenland coastline includes the southern part outside of the polar circle (latitude $> 67^\circ\text{N}$ )."

_EGUsphere, 2024_

## Author Comment (AC1)

**Response to the comments from reviewer 1**

**Summary**

In this paper, Zhang and coauthors present a modified Atmospheric River Scale tailored to analyze AR events in the polar regions. They first describe the justification for a polar AR scale by comparing the integrated water vapor transport (IVT) climatology in polar regions to the mid-latitudes, finding that most ARs impacting Greenland and Antarctica would go undetected by the original AR scale due to the colder and drier conditions in the polar regions. They introduce a modified version of the AR scale that includes three new "polar" categories with lower IVT thresholds, then use this scale to analyze the frequency, seasonality, and interannual trends in polar ARs. They assess the precipitation and melt impacts of the ARs identified by the new scale, finding that weak and moderate ARs account for most AR-related precipitation in Greenland and Antarctica, while stronger ARs are infrequent but cause extreme precipitation when they occur. Finally, they describe a web product that provides polar AR forecasts in real time.

The paper is well-organized and well-written overall. Most figures are clear and the references are extensive and appropriate. In my assessment the polar AR scale will be highly useful to a broad range of users, and I am impressed by how much information about ARs in Greenland and Antarctica can be captured by this simple but well-designed scale. However, I think there may be a serious error in the ice sheet melt analysis that should be addressed before the paper can be published, as described in my major comment below. I also have a number of minor comments and technical corrections, as described below.

We appreciate the valuable comments from reviewer 1.

Our responses are in blue below each comment from the reviewer.

**Major comments**

(1) I strongly suspect there is some error in the analysis of surface melt in Fig. 13. Numerous studies have documented an increase in Greenland Ice Sheet melt during the 21st century, but Fig. 13b shows more summer days with surface melt even in the 1980s compared with the 2000s. This is almost certainly incorrect. For a quick check, compare the years 1993 and 2012 using the Greenland Surface Melt Extent Interactive Chart at the National Snow and Ice Data Center Ice Sheets Today page (https://nsidc.org/ice-sheets-today/melt-data-tools). The NSIDC chart shows a much more extensive Greenland melt area throughout virtually all of JJA in 2012 compared to 1993, but Fig. 11b taken literally shows that 1993 was the most extensive melt year along the Greenland coast and had over 3 times the melt days of 2012. This figure also doesn't line up with Fig. 11b which shows increasing AR frequency along the Greenland coastline.

I am less familiar with melt trends in Antarctica so I am not sure if Figs. 13a and 13c contain obvious errors. The authors should check their analysis and the underlying microwave melt dataset in both Greenland and Antarctica for potential errors in processing.

[Figure]

*figure shared by reviewer*

We checked the calculation of surface melt in Fig.13 and found an issue caused by the missing data along the coastline of Greenland. Specifically, we used the grids along the coastline of Greenland (as shown in Fig.4c) for the analysis of Figs.13b&d, but the surface melt dataset has missing data in about one third of those grid cells on average during 1980-2022. Then we excluded the missing data, but the surface melt along the Greenland coastline still did not show a significant increase trend through the 21$^{st}$ century. The melt data from the limited number of coastal grid cells might introduce some uncertainties, and the result might be misleading.

Therefore, we re-calculated the surface melt for the entire Greenland using all grids with valid data instead of the coastline and re-created Figs.13b&d. In the new results, the surface melt in Greenland has a significant increase trend (+0.89 days per decade, P value = 0.002, the trend line was added in Fig.13b). The melt days in 2012 JJA is nearly doubled compared to the melt days in 1993 JJA. These are consistent with previous studies and the chart from NSIDC that the reviewer shared.

The Antarctica coastline has a similar missing data issue, although the impact seems not as large as for the Greenland coastline. To ensure reliability and keep consistency in Fig.13, we re-created Figs.13a&c to show the surface melt for Antarctica instead of the Antarctic coastline. In Antarctica, the surface melt data does not include the area higher than 1700 m of altitude, where melting is unlikely. The surface melt in Antarctica has a slight but not statistically significant decrease trend (-0.10 days per decade, P value = 0.135). This result is consistent with previous studies, which showed that there are no significant changes or slight decrease trend in surface melt in Antarctica.

The new Fig.13 shows that 23% and 26% of the surface melt is associated with ARs in Antarctica and Greenland respectively. We revised the text accordingly in Section 4.2 Surface melt and polar ARs (see tracked changes in the manuscript).

[Figure]

*Figure: The new Fig.13 in the manuscript. The surface melt was calculated in the entire Antarctica and Greenland instead of the coastlines.*

**Minor comments**

(1) General comment: Have the authors thought about adjustments to the polar AR scale that may need needed in future warming scenarios? Will the AR scale remain constant in climate change scenarios, despite the projected increases in IVT in the polar regions? Any new analysis on this topic is likely outside the scope of this manuscript, but it could be a nice addition to the paper to discuss this as a topic for future research in the conclusions section.

We agree that the future change of polar ARs is an important research topic, which is closely related to the future changes in the extreme weather trigged by ARs and the relevant surface mass balance in Antarctica and Greenland, but it is out of the scope of this manuscript as the reviewer mentioned. Following the reviewer's suggestion, we included some relevant discussion in Section 6. Conclusions and discussion (see tracked changes). Generally, both relative and fixed IVT thresholds were used to quantify the future changes of ARs in previous studies depending on different study goals. The AR scale is designed to objectively quantify the impact and strength of ARs based on their intensity and duration. Therefore, it is ideal to keep the IVT thresholds of the AR scale consistent under climate change. Because ARs with stronger IVT usually have higher impacts (e.g., more precipitation), and it is reasonable to rank them as a higher AR scale using consistent IVT thresholds in polar AR scale.

(2) Title: I suggest removing "CW3E" from the title, or at least stating the full name of the Center for Western Weather and Water Extremes in the title. The abbreviation "CW3E" will not be familiar to many in the cryospheric science readership of this journal. I also note that the Polar AR Scale is described as a collaborative effort between CW3E and the Byrd Center at Ohio State (L523–526), and the paper introducing the original scale (Ralph et al., 2019) does not describe it as the "CW3E Atmospheric River Scale".

We spelled out "CW3E" in the title, so the new title is "Extending the Center for Western Weather and Water Extremes (CW3E) Atmospheric River Scale to the Polar Regions". Both the regular AR scale introduced by Ralph et al. 2019 and the extended polar AR scale in this manuscript are led by CW3E. Therefore, we included CW3E in the title.

(3) L44: I suggest framing this sentence along the lines of "Our results show that that the polar AR scale better characterizes the strength and impacts of ARs in the Antarctic and Arctic regions than the original AR scale, and has the potential..."

We revised that sentence as suggested.

(4) L170–171: Why were 1-degree ERA5 data used instead of the finer native resolution of ERA5? Do the authors expect that this has any influence on their results? I note that the original AR Scale in Ralph et al. (2019) used 0.5-degree gridded data.

We use 1-degree ERA5 data in this manuscript since we are examining the AR scale in polar regions. The zonal length of a 1-degree grid cell in polar regions (e.g., ~38km at 70N/S) is roughly close to the zonal length of a 0.5-degree grid cell at middle latitudes (e.g., ~39km at 45N/S). Meanwhile, ARs are relatively large objects, which are usually a few hundreds to a couple of thousands of kilometers in size. Therefore, we think the 1-degree resolution is sufficient to examine the features of polar ARs. Another reason is that we planned to include a section to explore the uncertainties in the AR scale due to different reanalysis datasets and we used the 1-degree common grid. However, we did not include that part in the manuscript eventually due to the difference in IVT calculation. The IVT from ERA5 was calculated using model levels from the surface to the top of the atmosphere while the other reanalysis datasets only provide Q, U, and V at pressure levels. The uncertainties in the AR scale based on different reanalysis datasets might be caused by the different IVT calculation methods rather than the data itself. Therefore, we did not include that part in this manuscript.

A finer resolution (e.g., 0.25-degree) might capture a higher maximum IVT value in some extreme AR cases compared to the 1-degree resolution. However, given the large interval between IVT thresholds (50 kg/m/s for AR P1 – AR P3 and 250 kg/m/s for AR1 and higher ranks), using a finer resolution does not have a significant impact on the climatological results in this manuscript. Beyond this manuscript, different resolutions could be used accordingly in the polar AR scale framework for different research or applications (climate or weather, large-scale or local scale, etc.).

(5) L218–228: This is a nice analysis of the climatology of IVT in the parts of Greenland and Antarctica that extend outside of the polar latitudes.

Yes, it has a small impact on some of the results (e.g., Fig.4 and Fig.9) if we include the southern part of Greenland that extends outside of the polar latitudes. However, that narrow part extends to 60°N and is surrounded by a relatively warm ocean, which has a different climatology from the part within the polar region. Thus, we did not include that part in the main body of the manuscript. On the other hand, that the southern part is an important part of Greenland, so we repeated the analysis to include that part and showed the results in the Supplement. The impact from the small tip of the Antarctic Peninsula is ignorable, so we did not include the corresponding figures.

(6) L286–287, 319–321: Out of curiosity, do the authors know how many AR4 events there are in the historical record in Antarctica? I see in L456–457 that no AR5 events have ever been recorded in Antarctica, but it would be nice to state the number of AR4 events here to provide historical context for the March 2022 event. Would it be straightforward for the authors to include a map of the maximum AR category ever reached in the historical record at the Antarctic coastline points shown in Fig. 6d?

There was only one AR4 event identified during 1979-2022 in Antarctica, the extreme AR in March 2022 as shown in Fig.1 and Fig.6 (Fig.S3 in the revised version). It covered eight locations/grids along the coastline of the East Antarctica. We add a couple of sentences in Section 3.1 to clarify that.

This manuscript focuses on introducing the extended AR scale for polar regions and providing the relevant statistical results from a climatology perspective, which are usually less sensitive to the input data (e.g., the spatial and temporal resolution of the IVT data). However, the maximum AR scale in the historical record at a specific location has large uncertainties due to many factors, like the spatial and temporal resolution of the IVT data (e.g., 1.0 degree vs. 0.25 degree, 6-hourly vs. hourly), the calculation of IVT (integrated at model levels or limited number of pressure levels), or the different datasets (different reanalysis datasets or observation). Therefore, we did not include a figure showing the maximum AR scale events in the historical record at specific locations, which might include large uncertainties and be misleading.

We agree that a study focusing on the most extreme landfalling AR events along different locations of the Antarctic coast during the historical record is a good follow-up research.

(7) Figure 6d and elsewhere: How / why were the locations of the these points along the Antarctic coastline chosen to calculate AR scale data? Are they selected to be useful for particular communities, such as Antarctic research stations?

Basically, the dots in Fig.6d are located every 5 degrees in longitude along most parts of the Antarctic coastline. Meanwhile, there are additional locations along the coastline of the Antarctic Peninsula since it has a complex topography and coastline, and it is an area with more AR activities. In addition, we also included a dot at the Dome C station in the East Antarctic since it is an important observation site.

In this manuscript, we focus on the coastal regions of Antarctica and Greenland. The CW3E AR Scale Forecast tools for Antarctica described in Section 5 will include more locations according to research interests and application needs.

(8) Fig. 7: To help interpret these maps, it would help to add a few solid contours with contour labels. Perhaps the contours of 1, 5, and 10 average annual ARs could be labeled.

We used a different color map for both Fig.7 and Fig.8 so it is easier to read the AR frequency.

(9) Fig. 8: Why are there more AR 2 events (panel e) in this "Atlantic Arctic gateway" region than ARs in the weaker AR P1 through AR 1 categories (panels a–d)? Is this correct?

We checked the results in Fig.8 with a focus on the Atlantic Arctic gateway region and the results are correct. The Atlantic Arctic gateway region is quite different from the other regions along the north polar circle. It is the only wide and open ocean area along the north polar circle, and the extratropical cyclones (usually acting as a dynamical driver of ARs) are very active and relatively stronger over that region along the Atlantic storm track. As a result, the ARs over that region usually have a stronger IVT and a longer duration. Therefore, the ARs there tend to have relatively higher ranks with a maximum frequency in AR2.

(10) L385–389: Nice analysis of the seasonality of Greenland ARs. This is an interesting result and Fig. 10 is an interesting figure.

In addition to the current description and interpretation about the seasonality of ARs, we rewrote some of the discussion in the manuscript about the reasons for this seasonality (see the tracked changes in the manuscript).

(11) L445–456: How / why was this 12-hour window chosen to define AR-associated precipitation? Is there precedent for this method in the literature? I have not performed an extensive literature review but I note that Maclennan et al. (2022) defined AR-associated precipitation in Antarctica using precipitation from the time of the AR + the following 24 hours.

For a specific location, we considered precipitation under AR conditions, 12 hours before, and 12 hours after AR as the precipitation associated with ARs. This time window (from 12 hours before to 12 hours after AR) was selected because we found that precipitation occurs not only under AR conditions but also about 12 hours before and after AR, although the variability of the time window is large. Meanwhile, some previous studies (e.g., Maclennan et al., 2022 and Wille et al., 2021) defined the precipitation under AR conditions and within 24 hours after AR as the precipitation associated with ARs.

To better define the precipitation associated with ARs, we re-calculated the precipitation occurrence before and after AR conditions for the Antarctic and Greenland coasts. The results show that on average the precipitation rate after AR conditions decreases with time and the precipitation rate during the first 12 hours (0-12 hours after AR) is higher than the second 12 hours (12-24 hours after AR). This is consistent with Fig.1 from Maclennan et al., 2022. Meanwhile, we found that the precipitation rate in the 12 hours before AR is comparable to the second 12 hours (12-24 hours) after AR, but with a larger variability indicating the precipitation difference from case to case is larger in the 12 hours before AR.

Therefore, in the revised manuscript we decided to follow the previous studies (Maclennan et al., 2022; Wille et al., 2021) to define the precipitation under AR conditions and within 24 hours after AR conditions as the precipitation associated with ARs. The updated Fig.12 has very small differences from the old version (12 hours before and after AR). For example, the contributions of all ARs to the total precipitation are 32.0% and 32.2% for the Antarctic and Greenland coast, which are slightly different from the previous 32.4% and 31.8%. This is not surprising since most AR-related precipitation occurs under AR conditions and within 12 hours after that. We also revised the description and discussion in the manuscript accordingly.

(12) L484–485: This delay of 18–24 hours found by Mattingly et al. (2023) applies specifically to the delay between AR landfall in northwest Greenland and melt in northeast Greenland due to the foehn effect, not generally to all Greenland ARs.

Mattingly et al. (2023) found that there is a delay of 18–24 h between AR landfall in northwest Greenland and maximum foehn-induced melt in northeast Greenland, but the melt associated with ARs occurs during the period of -48 to +48 hours (mainly -24 to +48 hours) surrounding ARs as shown in their Fig.5c (copied below). In addition, Wille et al. (2019) found that after an AR makes landfall, the residual high precipitable water and resulting mixed-phase clouds continue to cause surface melt for around five days until the airmass is transported away in Antarctica. Therefore, if the melt occurs on the identified AR days or within one day before or two days after the AR days, the melt is classified as AR-related melt. We revised the relevant sentences to clarify that.

[Figure]

*Figure: Fig.5c from Mattingly et al. (2023), temporal evolution of foehn-driven melt in northeast Greenland in 500 m elevation bands during the −48 to +48 h period surrounding northwest Greenland ARs.*

(13) L546–548: Are there any plans to extend the CW3E polar AR scale forecasts to the Arctic, and to Greenland in particular? I could envision it being highly useful to the scientific and public communities in Greenland.

The CW3E polar AR scale forecast products for Greenland are in preparation. Similar to the Antarctic AR scale forecasts, we aim to provide AR Scale forecasts for Greenland based on dynamical model forecast data and the polar AR scale introduced in this manuscript. We added a couple of sentences at the end of Section 5 to mention that.

**Technical corrections**

- L35: application --> applications

Revised as suggested.

- L36: "the intensity"... of what? IVT?

It is the intensity of IVT. We revised that sentence to clarify.

- L38 and elsewhere (e.g. L568): Find a better word than "insufficient" to describe the unsuitability of the standard AR scale. I suggest "unsuitable". "Insufficient" implies that the scale does not reach high enough IVT values to characterize polar ARs, but the opposite is actually the case.

Here, we use "insufficient" to indicate that the ranks of the regular AR scale are not sufficient to cover the low-IVT polar ARs, so we need the extended ranks for polar ARs. We did not use "unsuitable" because the regular AR scale (AR1 – AR5) can still be used for polar ARs, and "unsuitable" may be misleading.

- L43: Antarctic --> Antarctica

Revised as suggested.

- L46: "observation, research, and forecasts" – this list is a grammatically incorrect mixture of singular and plural verbs. Please revise.

We revised it to "… to enhance communications across observation, research, and forecast for polar regions".

- L71: "the diabatic process" --> "diabatic heating"?

Revised as suggested.

- L75: "the polar ice" --> "the polar cryosphere"

Revised as suggested.

- L116: starts --> start

Revised as suggested.

- Fig. 1 caption: Labels b and c don't match the figure panels. They refer to panels c and b in the figure.

We corrected the caption of Fig.1 to match the labels b and c.

- L158: its --> their

Revised as suggested.

- L158, 525: The abbreviation "CW3E" is defined in multiple places in the manuscript.

We kept the full name of CW3E at the first place and removed the others.

- L170 and elsewhere (e.g. L177, L180): "data was" --> "data were". (The word "data" is a plural noun. Please check this throughout the manuscript.)

We revised it as suggested and also checked throughout the manuscript.

- L174: The abbreviation "EA" is not defined anywhere in the manuscript.

We changed "EA" to "East Antarctica".

- L191, L196: The phrases "southern hemisphere" and "northern hemisphere" are not capitalized in this paragraph, but "Southern Hemisphere" and "Northern Hemisphere" are capitalized elsewhere in the manuscript (e.g. L204–205, L397). Please be consist with capitalization.

We used "Southern Hemisphere" and "Northern Hemisphere" through the manuscript.

- L199: A space is needed before the opening parenthesis in "(Fig. 3b)".

We added the space there.

- L204: The caption states that the maps show the Southern and Northern Hemisphere, but technically the maps only show the mid- and high-latitude areas of each hemisphere.

We changed it to "… the middle and high latitudes of the Southern Hemisphere (a) and Northern Hemisphere (b) …" in the caption.

- L212: percentages --> percentiles

Revised as suggested.

- L253: What are "variant" meteorological conditions? Please rephrase.

We rephrased that to "… associated with different meteorological conditions, such as the precipitation amount and rate."

- L330: Rather than "the gap between Greenland and Northern Europe", a more specific term that is often used to describe this region in the atmospheric and marine science literature is the "Atlantic gateway to the Arctic", or it could also be described as the "Nordic Seas".

We used "Atlantic gateway to the Arctic" instead of "the gap between Greenland and Northern Europe" as suggested.

- L426: increase --> increasing

Revised as suggested.

- L555: An open parenthesis is missing before the word "colored"

We added the open parenthesis.

- L603: was --> were

Revised as suggested.

- L608: illustrating --> illustrative

Revised as suggested.

---

## Author Comment (AC2)

**Response to the comments from reviewer 2**

**Summary**

This paper presents a novel tool for identifying polar atmospheric river (AR) events and their intensities. The methodology builds on Ralph et al. (2019)'s mid-latitude AR scale by including three new categories for polar ARs, which account for lower amounts of atmospheric moisture at higher latitudes. The authors present the methodology and results from the new scale, with analysis of the frequency of ARs of the different rankings in each polar region, and then discuss the precipitation and surface melt in Greenland and Antarctica attributed to ARs detected by this scale. Furthermore, the authors present a new tool for forecasting ARs and AR intensity in Antarctica. This represents a significant step forward in the development of a wider range of tools to detect and analyze polar ARs, and because this tool is Eulerian (i.e., the AR ranking is on a point-by-point basis), it allows for novel interpretations of the frequency and likelihood (risk) of ARs of different intensities.

The paper is generally well-organized and written in a clear and effective manner. However, I have several major comments below, and overall, my primary concern is that the results and analyses presented in this paper are not well grounded in the results from previous studies. There is little comparison of the AR frequency and precipitation and melt impacts found here to prior studies of these features (using different AR detection algorithms) in both Greenland and Antarctica. I think this paper could be significantly improved by providing more context and comparison to the results from previous studies in both polar regions, regarding both the method for detection and the impacts attribution.

We appreciate the valuable comments from reviewer 2.

Our responses are in blue below each comment from the reviewer.

**MAJOR COMMENTS**

(1) The use of integrated vapor transport (IVT) versus the meridional component of integrated vapor transport (vIVT) for AR detection in the polar regions – Figure 1 in Shields et al., 2022 (https://agupubs.onlinelibrary.wiley.com/doi/full/10.1029/2022GL099577) shows a comparison between the Wille et al., 2021 AR detection algorithm, which is based on vIVT, and a number of standard global AR detection algorithms, most of which use IVT for AR detection. The figure highlights that IVT-based detection methods struggle to capture Antarctic ARs, particularly in the interior of the ice sheet. I think it would be incredibly important to assess the limitations of using one or the other (IVT or vIVT) in detecting ARs in the polar regions in this paper, given the results from previous studies like Wille et al. 2021. For example, in the context of this result presented in Shields et al. 2022, what does a frequency map of ARs detected by this new AR scale look like compared to the Wille et al. 2021 algorithm (as in, a frequency difference map)? Or alternatively, how well does the AR scale capture known AR events, for example those presented in Gorodetskaya et al. (2014)?

There are many different AR detection tools (ARDTs), including many regular global ARDTs and some ARDTs specifically designed for Antarctica (Wille et al. 2019, 2021). Shields et al. (2022) compared the regular global ARDTs and the Antarctic ARDTs, and they found that the Antarctica ARDT from Wille et al. 2021 can capture more ARs in the interior of the ice sheet

(Fig.1 in Shields et al., 2022). This is likely because the ARDT from Wille et al. 2021 uses only the meridional component of IVT (vIVT) and uses the 98[th] percentile as the threshold for vIVT to identify ARs. This allows AR detection over the dry Antarctic interior, where IVT is usually extremely low. We agree that the meridional component of water vapor transport (vIVT) is critical for moisture intrusions into the Antarctic interior. The Antarctic ARDT using vIVT from Wille et al. 2021 can successfully capture the Antarctic interior ARs as shown in previous studies (e.g., Wille et al., 2021; Shields et al., 2022).

For the polar AR scale in this manuscript, we believe IVT is the most suitable variable. Because if we use only vIVT instead of IVT, many zonal ARs (zonal IVT plays a dominant role) in Greenland and some parts of Antarctica may be missed due to their relatively weak vIVT (e.g., the landfalling ARs in the figure below). On the other hand, when a potential AR case has a strong vIVT, it is very likely to have a strong total IVT, so the polar AR scale using 100 kg/m/s as a minimum IVT threshold usually can capture them over the ocean and the coastal area in polar regions. For example, the two landfalling ARs over East Antarctica on 19 May 2009 and 15 February 2011, presented in Gorodetskaya et al. (2014) and mentioned in the reviewer's comment, are captured by the polar AR scale and ranked as AR2 and AR1 respectively.

For the Antarctic interior, IVT is always extremely low due to the low temperature and high altitude. The climatological mean IVT in most Antarctic interior is around or below 10 kg/m/s according to ERA5 (Fig.3). ARs usually decay very quickly after landfalling in the Antarctica interior. As a result, the AR frequency in the Antarctic interior based on the polar AR scale is very low (Fig.7). We tested adding another low-IVT threshold (50 kg/m/s) in the polar AR scale, it could capture slightly more AR events in the Antarctic interior; however, it introduced a lot of false alarms over the coast and ocean in both polar regions since the climatological mean IVT there is nearly or around 50 kg/m/s. Therefore, we used 100 kg/m/s as the IVT minimum threshold in the polar AR scale.

In addition, the AR scale is mainly designed to objectively quantify the strength and impact of AR events, which is different from the regular ARDTs as mentioned in the manuscript. Although the AR scale can be used to identify AR conditions independently, it can also be used with a regular ARDT. For example, Guan et al. 2023 (*Global Application of the Atmospheric River Scale*, JGR-Atmosphere) used an ARDT to identify ARs first and then used AR scale to quantify the strength of those identified ARs.

We added a brief relevant discussion (IVT and vIVT, ARs in Antarctica interior, etc.) in the second paragraph of Section 6, Conclusions and Discussion.

[Figure]

*Figure: right panel is Fig.2d in this manuscript showing a zonal AR landfalling in Greenland in November 2021; and left panel is Fig.4c from Zou et al. (2023) showing a zonal landfalling AR over the Antarctica Peninsula in December 2018.*

(2) Description of how to determine the rank of an AR event – regarding the ranking of AR events in the polar regions (as described on P11 L246 to L268), did the authors consider adjusting the duration requirements for ARs as well? In the Wille et al. (2021) AR detection algorithm, there are frequently ARs that appear to make landfall for only a few hours, less than the amount of time required to meet the 24-hour qualification presented here. I found the discussion on the IVT climatology analysis and the choice of IVT thresholds for the polar scale quite interesting, but I am wondering if you also examined the sensitivity of the AR scale detection method to the time period requirements to meet certain rankings?

At a specific location, the maximum IVT of an AR case is used to determine the initial rank of the polar AR scale according to the IVT thresholds of each rank; then the duration of the AR condition is used to adjust the initial rank and determine the final AR scale rank (see details in Section 2.3). Specifically, if the AR duration exceeds 48 hours, promote the scale by 1 rank; if the AR duration is less than 24 hours, demote the scale by 1 rank. In the polar AR scale, we did not make any adjustments to the duration thresholds.

We added three new low-IVT thresholds (100, 150, and 200 kg/m/s) in the polar ARs scale because of the low-IVT climatology in polar regions. Consequently, many low-IVT polar ARs can be captured and ranked by the polar AR scale. Without these new low-IVT thresholds, the AR scale will miss the majority of polar ARs.

Regarding the duration of ARs, the two thresholds (24 and 48 hours) are used to adjust the initial rank (promote or demote the scale by 1 rank). Because given the same intensity (IVT), when the duration of an AR is longer, it tends to have a stronger impact (e.g., more precipitation); and vice versa. In other words, if an AR makes landfall for only a few hours, the rank of its scale will only be demoted by 1 rank (e.g., demoted from AR P3 to AR P2) due to its short duration and thus relatively weak impact. Only when an AR's initial rank is AR P1 (maximum IVT is 100-150 kg/m/s) and its duration is a few hours, it will not be identified as a polar AR due to its weak IVT and short duration, which is unlikely to have a significant impact. In addition, if we lower the 24-hour duration threshold, it may introduce many weak signals (relatively weak IVT existing for a very short period) that are not real ARs. Therefore, we did not change the duration thresholds.

We added a couple of sentences to discuss AR durations in Section 2.3.

(3) Also regarding the AR scale description (now P12 L276 to 290), I'm not sure that the March 2022 AR-heatwave event is optimal in showcasing the capabilities of the new polar AR scale, since it ranks as an AR4 on the midlatitude scale. While this was a standout event, and it's interesting to know how it ranks, I think it could be helpful to provide greater detail on known events that rank between AR P1 and ARP3, given that this will be the most relevant application for this scale. I am aware that an example of this was mentioned earlier in the text in Figure 2, as well as a brief description from L291 to L294 on P13. I would strongly encourage the authors to provide details on ranking these types of events in the level of detail presented for the March 2022 event.

We agree that the extreme AR in March 2022 may not be the best case to demonstrate how to use the polar scale since it is ranked as AR4, not in the ranks of the new polar AR scale. Instead of that, we created a new Fig.6 to use an AR case in December 2022 (the Antarctica AR in Fig.2), which is ranked as AR P3 (a rank of the new polar AR scale). Meanwhile, we moved the previous Fig.6 to Supplement since it is also a good example to show how to use the polar AR scale and how to adjust the initial AR scale based on its duration. We revised the corresponding text in the manuscript.

(4) Seasonality of polar ARs – I would ask that the authors please include the statistical significance of the results on AR seasonality on P17 – P18. Regarding the comparison of the AR scale results with the Wille et al. (2021) findings on AR seasonality, the authors suggest that the Wille algorithm has a higher frequency of ARs in winter months because the vIVT thresholding method (instead of IVT) is more closely tied to extratropical cyclones. My interpretation of the seasonality difference is that the Wille algorithm uses a threshold for vIVT that accounts for the seasonality of vIVT, where the vIVT threshold is higher in summer, due to higher atmospheric temperatures and increased moisture in the atmosphere, and lower in the winter, when conditions are drier. Comparatively, the polar AR scale uses an absolute threshold for ARs regardless of the season. Because of this absolute threshold, I would certainty expect the polar AR scale to detect more ARs in summer than in winter, just based on seasonal differences in the amount of atmospheric moisture / IVT / vIVT. To me, this doesn't suggest that there can be a conclusion formed about which detection method is more or less affected by seasonality in extratropical cyclones. I would be interested to hear what the authors think about this and how it relates to the seasonality analysis. (and as a sidenote, can the authors please also provide a citation for "is closely related to the occurrence of extratropical cyclones, which are more active during JJA in the Southern Hemisphere.")

We agree with the reviewer's interpretation of the comparison in seasonality between this study and Wille et al. (2021). In the AR detection method from Wille et al. (2021), the vIVT threshold is a relative threshold (the 98th percentile), which accounts for the seasonality of vIVT. Because their vIVT threshold is higher in summer due to the higher temperature and more moisture and is lower in winter due to the low temperature and less moisture in the atmosphere. In contrast, the polar scale introduced in this manuscript uses a fixed IVT minimum threshold (100 kg/m/s). As a result, it identifies more ARs in summer due to the higher temperature/moisture and fewer ARs in winter due to the lower temperature/moisture. That is the main reason for the difference in seasonality between this study and Wille et al. (2021). We rewrote that part in Section 3.2 "Seasonality of Polar ARs" based on the discussion above.

We added the spread (standard deviation) of the total AR frequency for each month in Fig. 10.

(5) Precipitation analysis for polar ARs – could the authors please describe what the basis is for using the 12 hour before and after window for AR-attributed precipitation (mentioned on P20 L444)? I would be interested to know if the authors used a method to determine when AR precipitation tends to fall with respect to the timing of landfall (how long before, during, and after), given that this really impacts how much precipitation we attribute to ARs and their relative importance in contributing to the surface mass balance of the Greenland and Antarctic

ice sheets. Or alternatively, is this window based on a previous study of AR precipitation in polar regions?

For a specific location, we considered precipitation under AR conditions, 12 hours before, and 12 hours after AR as the precipitation associated with ARs. This time window (from 12 hours before to 12 hours after AR) was selected because we found that precipitation occurs not only under AR conditions but also about 12 hours before and after AR, although the variability of the time window is large. Meanwhile, some previous studies (e.g., Maclennan et al., 2022 and Wille et al., 2021) defined the precipitation under AR conditions and within 24 hours after AR as the precipitation associated with ARs.

To better define the precipitation associated with ARs, we re-calculated the precipitation occurrence before and after AR conditions for the Antarctic and Greenland coasts. The results show that on average the precipitation rate after AR conditions decreases with time and the precipitation rate during the first 12 hours (0-12 hours after AR) is higher than the second 12 hours (12-24 hours after AR). This is consistent with Fig.1 from Maclennan et al., 2022. Meanwhile, we found that the precipitation rate in the 12 hours before AR is comparable to the second 12 hours (12-24 hours) after AR, but with a larger variability indicating the precipitation difference from case to case is larger in the 12 hours before AR.

Therefore, in the revised manuscript we decided to follow the previous studies (Maclennan et al., 2022; Wille et al., 2021) to define the precipitation under AR conditions and within 24 hours after AR conditions as the precipitation associated with ARs. The updated Fig.12 has very small differences from the old version (12 hours before and after AR). For example, the contributions of all ARs to the total precipitation are 32.0% and 32.2% for the Antarctic and Greenland coast, which are slightly different from the previous 32.4% and 31.8%. This is not surprising since most AR-related precipitation occurs under AR conditions and within 12 hours after that. We also revised the description and discussion in the manuscript accordingly.

(6) Surface melting time series in Figure 13 – I am surprised to see a relatively low number of melt days in 2007 in Greenland compared to the 1990s, when we know 2007 was a record melt year for Greenland (Mote et al. 2007 - https://agupubs.onlinelibrary.wiley.com/doi/full/10.1029/2007GL031976). Similarly, I would expect anomalously high melt in Antarctica for the 2019/2020 austral summer. Could the authors double-check the melt analysis presented in this figure, or compare the melt observations used with another melt dataset? I am not an expert on surface melting, but the Greenland melt time series especially does not necessarily look how I would expect it to. Also, when describing the satellite-observed melt in the methods, both papers cited (Picard and Fily, 2006, and Torinesi et al., 2003) are Antarctic – is there a study you can cite that applies these observations to Greenland as well?

We checked the calculation of surface melt in Fig.13 and found an issue caused by the missing data along the coastline of Greenland. Specifically, we used the grids along the coastline of Greenland (as shown in Fig.4c) for the analysis of Figs.13b&d, but the surface melt dataset has missing data in about one third of those grid cells on average during 1980-2022. Then we excluded the missing data, but the surface melt along the Greenland coastline still did not show a significant increase trend through the 21$^{st}$ century. The melt data from the limited number of coastal grid cells might introduce some uncertainties, and the result might be misleading.

Therefore, we re-calculated the surface melt for the entire Greenland using all grids with valid data instead of the coastline and re-created Figs.13b&d. In the new results, the surface melt in Greenland has a significant increase trend (+0.89 days per decade, P value = 0.002, the trend line was added in Fig.13b). The 2007 JJA in Greenland is one of the top three years with the most surface melt days by 2007, which is consistent with the results from Mote et al. 2007.

The Antarctica coastline has a similar missing data issue, although the impact seems not as large as for the Greenland coastline. To ensure reliability and keep consistency in Fig.13, we re-created Figs.13a&c to show the surface melt for Antarctica instead of the Antarctic coastline. In Antarctica, the surface melt data does not include the area higher than 1700 m of altitude, where melting is unlikely. The surface melt in Antarctica has a slight but not statistically significant decrease trend (-0.10 days per decade, P value = 0.135). This result is consistent with previous studies, which showed that there are no significant changes or slight decrease trend in surface melt in Antarctica. Meanwhile, the 2019/2020 DJF has a relatively high number of melt days in the new result.

The new Fig.13 shows that 23% and 26% of the surface melt is associated with ARs in Antarctica and Greenland respectively. We revised the text accordingly in Section 4.2 Surface melt and polar ARs (see tracked changes in the manuscript). We also added a reference (Colosio et al. 2021) which used a similar surface melt data of Greenland (a very similar algorithm and raw input data).

[Figure]

*Figure: The new Fig.13 in the manuscript. The surface melt was calculated in the entire Antarctica and Greenland instead of the coastlines.*

(7) Role of IVT intensity in relation to AR impacts (as described on P27 L593) – "but including an objective description of AR strength can improve the understanding of ARs' impacts on polar regions". Baiman et al., 2023 (https://agupubs.onlinelibrary.wiley.com/doi/full/10.1029/2022JD037859) recently showed that the strength of the AR is not necessarily correlated with precipitation impacts, focusing on the region of Dronning Maud Land in Antarctica. The study found that one of the more important factors in determining AR precipitation intensity is having a mechanism for lift to produce the precipitation. I think it would be valuable to mention this in the discussion section and how it relates to the AR ranking – precipitation impacts you found in this study.

We agree that the IVT intensity is not the only factor controlling the precipitation during an AR event. The precipitation is usually determined by both the water vapor transport (IVT) and the lift of the air mass (so condensation). Generally, the lift mechanism could be related to some dynamic weather systems (e.g., warm conveyor belts of extratropical cyclones) or topography (topographic lifting). However, the lift mechanism varies greatly from case to case due to the different large-scale circulation, mesoscale convection, geophysical locations, etc. Even for the same topography, the different orientations of IVT may cause a big difference in lifting and then precipitation. In addition, the lift mechanism of an AR may change through its life cycle. For example, a warm conveyor belt of an extratropical cyclone may play a dominant role in lifting when an AR moves over the ocean and then the terrain may become a main factor for lifting when the AR makes a landfall and strikes an inland mountain.

However, the polar AR scale aims to objectively quantify the strength and impact of an AR itself at a specific location. Assuming that the lift mechanism is the same, the AR scale (determined based on IVT and duration of the AR) is closely correlated to precipitation. The other factors contributing to precipitation are not beyond the scope of the AR scale. We have added a couple of sentences to discuss that in Section 6.

**MINOR COMMENTS**

Title – I'm not sure how many cryospheric scientists are familiar with CW3E (the acronym), so I would recommend spelling it out in the title or removing it

We spelled out "CW3E" in the title, so the new title is "Extending the Center for Western Weather and Water Extremes (CW3E) Atmospheric River Scale to the Polar Regions".

P3 L75 – "notable influence on the polar ice" is not the clearest word choice/descriptor for ARs impacts on surface mass balance and ice shelf stability

We changed it to "important influence on the polar ice".

P3 L76 to L81 – I find it misleading that this section on AR impacts begins with "hot spells and heatwaves" and only includes "as well as intense snow accumulation" at the end. Many of the studies cited in this section have shown that by far the dominant impact of ARs is snowfall,

especially in Antarctica. I would recommend that the authors frame the impacts description to reflect the relative importance of each AR impact in the present – though of course with the caveat that this might change in a warming climate (as mentioned on P4 L109).

We rewrote those sentences to describe the importance of the AR impacts in the present first and then mentioned the caveat that these might change in a warming climate (see the tracked changes in the manuscript).

P4 L89 – "ARs can also interact with other weather systems" – by interact, do you mean compound? Or as AR-extratropical cyclone systems?

We mean "compound", and we changed it to "ARs can also compound with other weather systems, …".

P4 L90 to L107 – this reads more like a detailed summary of the heatwave than an introductory paragraph – consider condensing or tie more directly to the motivation for this study

We used this extreme AR case in March 2022 to demonstrate the great impacts of ARs in polar regions. We agree that this part could be shortened, and we trimmed it in the manuscript.

P4 L107 – "Under a warming climate, the extreme ARs are expected to increase in both frequency and intensity" – I think the studies cited here all refer to the midlatitudes, is that correct? If so, I would recommend mentioning that in the sentence, or alternatively looking for polar/Arctic-specific studies to cite (I don't know that this has been done for ARs in the Antarctic….).

We clarified that the previous studies about AR's future changes mainly focused on the middle latitudes in that sentence.

P5 L126 – "using flexible thresholds" – it would be interesting and helpful here to have one or two sentences that elaborate on what a flexible threshold means (as in, a percentile of vIVT or IVT relative to the climatology, etc.), as well as why a different (lower) threshold is needed for polar ARs.

Here "using flexible thresholds" means using different fields (e.g., IWV, vIVT, and IVT) and different thresholds (85th percentile, 98th percentile, etc.). Those methods identify ARs as objects in space with AR features using flexible thresholds, and they have their advantages for different research and application goals. Meanwhile, as we mentioned in the following sentence, "Ralph et al. (2019) suggested that it is also useful to identify ARs from an Eulerian perspective (defining an AR as a sequence of relevant meteorological conditions at a specific location of interest)", which is a different framework. The lower IVT thresholds are needed in the polar AR scale due to the substantially lower temperature and moisture in polar regions. We added two sentences to clarify.

P5 L129 – "on-the-ground applications and communications". I'm not sure what "communications" refers to here – as in, communicating to meteorologists and fieldworkers at weather stations to collect observations during the AR period?

In this study, we introduced the polar AR scale as an objective framework to quantify the strength and impact of ARs in polar regions based on both the duration of the AR condition and the maximum IVT at a specific location. This polar AR scale could be used by the meteorologists, fieldworkers, and anyone else interested in polar ARs to quantitatively describe or rank the strength of polar ARs. The regular AR scale introduced by Ralph et al. 2019 is widely used in research, forecasts, and public media reports to rank ARs at middle latitudes. The polar AR scale is designed for the same purpose but specifically for polar ARs.

Figure 2 – I would strongly recommend labelling the Ross Ice Shelf on the Antarctic maps, since scientists less familiar with the geography of the Antarctic continent may not know where it is. Similarly, it may be helpful to label East Greenland as well.

We labeled the Ross Ice Shelf on the Antarctic maps and the East Greenland on the Greenland map in Fig. 2 as suggested.

P7 L170 – what was the motivation for decreasing the resolution of the ERA5 data from 0.25 x 0.25 deg at hourly resolution to 1 x 1 deg at 6-hourly resolution? (is there a reason not to use the data with higher resolution?)

We use 1-degree ERA5 data in this manuscript since we are examining the AR scale in polar regions. The zonal length of a 1-degree grid cell in polar regions (e.g., ~38km at 70N/S) is roughly close to the zonal length of a 0.5-degree grid cell at middle latitudes (e.g., ~39km at 45N/S). Meanwhile, ARs are relatively large objects, which are usually a few hundred to a couple of thousands of kilometers in size. Therefore, we think the 1-degree resolution is sufficient to examine the features of polar ARs.

Another reason is that we planned to include a section to explore the uncertainties in the polar AR scale due to different reanalysis datasets, so we used the 1-degree common grid. However, we did not include that part in the manuscript eventually due to the difference in IVT calculation. The IVT from ERA5 was calculated using model levels from the surface to the top of the atmosphere while the other reanalysis datasets only provide Q, U, and V at pressure levels. The uncertainties in the AR scale based on different reanalysis datasets might be caused by the different IVT calculation methods rather than the data itself. Therefore, we did not include that part in the manuscript.

P9 L208 – I would specify that you are including Antarctic ice shelves in the coastline

We revised that sentence to clarify it as suggested.

P9 L212 – "percentages of IVT" as in percentages of IVT "values"?

Yes, it is the percentage of IVT samples (values). We have revised that sentence to clarify.

P9 L218 – I find it slightly surprising that the authors decided to exclude southern Greenland from the analysis, given that surface melting, an impact highlighted by in the introduction, is prominent in this region. From the ice sheet and sea level rise perspective, it seems valuable to include the whole of Greenland in this study, especially since the new AR scale still includes the midlatitude scale for AR intensity.

The southern Greenland part extends to 60°N. It is narrow and surrounded by a relatively warm ocean, where the climatology is quite different from the other parts of Greenland within the polar region. The extended polar AR scale is designed for the ARs within the polar regions. If we include the southern Greenland part, it will introduce many additional AR events outside the polar region (comparing Fig.9b and Fig.S2b) since even the climatological mean IVT is very close to 100 kg/m/s (the minimum IVT threshold for AR P1) along the coast of the southern Greenland (Fig.3b). Thus, we excluded the narrow southern Greenland part to focus on the ARs within polar regions in the main body of this manuscript.

On the other hand, we agree that the southern part is an important part of Greenland as the reviewer mentioned, so we repeated the analysis to include that part and showed some of the results (IVT distribution and AR frequency along the coastline) in the Supplement.

P14 L306 to L312 – the distance covered by the Antarctic coastline is huge, so I would recommend being more specific in the locations described here (i.e., instead of "most of the East Antarctic coast", mention the names of specific regions).

We rewrote that sentence to describe the AR frequency along the Antarctic coast more specifically as suggested.

P14 L311 – I'm confused by this sentence: "… there are more AR P1 events over the inland area close to the coast compared to a similar area in East Antarctica."

We deleted that part of the sentence since it is redundant and confusing.

P14 L314 to L319 – the analysis jumps from AR P1 to AR1 rankings – since AR P2 and P3 are the other two new categories, could you list the statistics for these events here too?

We added a couple of sentences to describe the frequency of AR P2 and P3 in Fig.7 and Fig.8.

Figures 7 and 8: the red-white color map used in this figure makes it extremely difficult to discern any difference between AR frequencies from AR P1 to AR P3, and the colors are very washed out for AR3 and AR4. Can you try using a different color map for these figures that better highlights spatial differences among the panels? Also, please write out the full figure caption for Figure 8 instead of referring to Figure 7.

We used a different color map, so it is easier to read the AR frequency in Fig.7 and Fig.8.

We also wrote out the full caption for Fig.8.

P16 L354 – "0.011 events" and "0.001" are numbers that I find slightly difficult to interpret in a physically meaningful way – would it be possible to list the number of AR3 and AR4 events that occurred in parentheses?

We added a sentence to clarify the total number of AR4 during this historical period (1979-2022) in Antarctica. There was only one AR4 event, the landfalling AR4 over East Antarctica in March 2022, according to the 1-degree and 6-hourly ERA5 IVT. It covered 8 grid cells along the Antarctic coast. We did not emphasize much the exact number of extreme AR events (like AR3 and AR4) since the number might be sensitive to the spatial and temporal resolution of the IVT data, or even to the datasets (e.g., different reanalysis datasets). We used the average number of AR events per year per location (like 0.011 events for AR3) to emphasize the rareness of those events from a climatological perspective. We revised a couple of sentences in the manuscript to clarify that.

P19 L408 – "along Greenland" --> "along the Greenland coast?"

Revised as suggested.

P19 L424 – "coast of East Greenland" – this does not include southern Greenland, right? If so, would recommend saying "central-north East Greenland coast"

We agree that "central-north East Greenland coast" is a more accurate name, and have changed it through the manuscript.

P21 L447 – what is the standard deviation in annual AR precipitation?

The vertical bars in Figs.12a&b are ±1 standard deviation in the contribution of annual AR precipitation. We added the note in the caption of Fig.12 and added one sentence to describe it in the manuscript.

P21 L466 to L469 – nice summary!

Thank you!

P24 L522 – is there a citation you can include for YOPP-SH?

We added Bromwich et al. 2024 here as a reference for YOPP-SH. That paper was submitted to BAMS and is in revision now. It provides a detailed summary for the latest YOPP-SH campaign.

P25 L527 – I am missing a methods/data description with respect to GEFS, which is introduced here. It sounds like this might be described more in Bromwich et al. 2024 (is this in review?), but I think it would be highly relevant to include more information on the reliability of the GEFS in capturing the intensity, extent, and duration of Antarctic ARs.

We agree that it is important to evaluate the reliability of the GEFS in capturing the intensity, extent, and duration of Antarctic ARs before using the relevant forecast products, including the CW3E Antarctic AR forecast tools. However, this is beyond the scope of this manuscript. It could be a good follow-up study. The paper of Bromwich et al. 2024 submitted to BAMS is in revision now and it introduced some details about how the CW3E Antarctica AR forecast tools were used in the YOPP-SH campaign. We added a sentence to clarify that.

P26 L565 – you introduced the AR acronym quite a bit earlier in the paper

We removed that and used only "AR" there.

P27 L587 – "related to" as in "associated with"?

We changed it to "associated with".

P28 L613 – "… enhance situational awareness, contributing to timely preparedness and effective decision-making for high impact events…" I'm not sure who this is referring to – citizens, fieldworkers, meteorologists who can launch weather balloons during the events? I'm not aware of structures outside of the YOPP-SH campaign (and maybe the research stations that need to keep fieldworkers safe?) that employ decision making strategies for polar AR events, so I would welcome more specificity/clarity here on what this sentence means.

We are referring to the fieldworkers and meteorologists who are working in polar regions and/or are interested in the extreme weather associated with polar ARs. Meanwhile, from the climate perspective, ARs' impacts on the surface melt in Antarctica and Greenland are critical under climate change. We rewrote that sentence to be more specific in the manuscript.

---

## Author Response (AR2)

**Reviewer 1 – Second Round**

Thanks to the authors for their thorough and thoughtful responses to mine and the other reviewer's comments. I request a few more technical corrections listed below. After these minor corrections, I feel this paper will be suitable for publication.

We appreciate the thoughtful suggestions from reviewer 1.

Our responses are in blue below each comment from the reviewer.

**Technical corrections**
- L45–46: "observation, research, and forecast communities" --> "observational, research, and forecasting communities"

Revised as suggested.

- L97: "vertical-integrated" --> "vertically-integrated"

Revised as suggested.

- L130–131: "...to better capture the ARs less well-structured" – What does this mean?

We have revised it to "… to better capture ARs that are less well-structured."

- L192: "was interpolated" --> "were interpolated"

Revised as suggested.

- L195: "data is" --> "data are"

Revised as suggested.

- L357: "stronger" --> "strong"

Revised as suggested.

- L368: "higher" --> "high"

Revised as suggested.

- L410: "weaker" --> "weak"

Revised as suggested.

- L425–427: This sentence is an incomplete sentence fragment. Please revise.

We have revised this sentence: "Their vIVT threshold is higher in summer due to the increased temperature and moisture, and lower in winter because of the decreased temperature and moisture."

- L516: "data is" --> "data are"

Revised as suggested.

- L532–533: "AR's contribution to surface melt" --> "AR contributions to surface melt"

Revised as suggested.

- L537: "AR's" --> "AR"

Revised as suggested.

- L581: "ranking" --> "rankings"

Revised as suggested.

- L635–636: "polar AR scale" --> "the polar AR scale"

Revised as suggested.

- L649: "AR's contribution" --> "AR contributions"

Revised as suggested.